There are amendments to this paper

# Revisiting the pH-gated conformational switch on the activities of HisKA-family histidine kinases

Cristina Mideros-Mora [1,2], Laura Miguel-Romero [1,6], Alonso Felipe-Ruiz [1], Patricia Casino [3,4,5]* & Alberto Marina [1,5]*

Histidine is a versatile residue playing key roles in enzyme catalysis thanks to the chemistry of its imidazole group that can serve as nucleophile, general acid or base depending on its protonation state. In bacteria, signal transduction relies on two-component systems (TCS) which comprise a sensor histidine kinase (HK) containing a phosphorylatable catalytic His with phosphotransfer and phosphatase activities over an effector response regulator. Recently, a pH-gated model has been postulated to regulate the phosphatase activity of HisKA HKs based on the pH-dependent rotamer switch of the phosphorylatable His. Here, we have revisited this model from a structural and functional perspective on HK853–RR468 and EnvZ–OmpR TCS, the prototypical HisKA HKs. We have found that the rotamer of His is not influenced by the environmental pH, ruling out a pH-gated model and confirming that the chemistry of the His is responsible for the decrease in the phosphatase activity at acidic pH.

[1] Instituto de Biomedicina de Valencia, Consejo Superior de Investigaciones Científicas (IBV-CSIC), Jaume Roig 11, 46010 Valencia, Spain. [2] Universidad UTE, Facultad de Ciencias de la Salud Eugenio Espejo, Rumipamba s/n, Quito, Ecuador. [3] Departament de Bioquímica i Biología molecular, Universitat de València, Dr. Moliner 50, 46100 Burjassot, Spain. [4] Estructura de Recerca Interdisciplinar en Biotecnologia i Biomedicina (ERI BIOTECMED), Universitat de València, Dr Moliner 50, 46100 Burjassot, Spain. [5] CIBER de enfermedades raras (CIBERER-ISCIII), Madrid, Spain. [6] Present address: Institute of Infection, Inmmunity and Inflammation, University of Glasgow, Glasgow G12 8TA, UK. *email: patricia.casino@uv.es; amarina@ibv.csic.es

Two-component systems (TCS) are the main signaling mechanism used for bacteria to sense environmental changes and respond in order to adapt and survive[1]. TCS are composed of two main players, a sensor protein called histidine kinase (HK) and an effector protein called response regulator (RR)[2]. Prototypical HKs are membrane bound homodimers containing an extracellular sensor domain of variable architecture and a cytoplasmic conserved catalytic machinery[1,2]. The HK catalytic core is comprised of an ATP-binding domain (CA domain) and a phosphoacceptor domain containing the phosphorylatable His residue (DHp domain)[3]. DHp domain has been used to classify HKs in different families representing HisKA the dominant family with ~80% of known HKs[4,5]. In turn, RRs tend to work as transcriptional factors regulating expression of specific genes. RR activity is controlled by the phosphorylation of a conserved Asp residue located at the N-terminal receiver domain (REC domain)[3]. Signaling by TCS is the result of three different reactions;[6] an initial kinase reaction which corresponds to the autophosphorylation of the HK in a conserved His residue on the DHp domain. In this reaction, the CA domain recruits the ATP molecule and its γ-P is attacked by the DHp phosphoacceptor His, which is activated by a conserved adjacent acidic residue[6,7]; the kinase reaction has been visualized in the crystal structures of two HisKA HKs, CpxA, and a chimeric version of EnvZ, both trapped catalyzing this reaction where the His acts as a nucleophile to attack the γ-phosphate of ATP helped by its interaction with a neighbored conserved acidic residue that acts as a general base to increase the His nucleophilic character[7,8]. Secondly, the phosphoryl group is transferred from the His to the conserved Asp residue at the REC domain in the phosphotransfer reaction[6]. This reaction has not been visualized yet for the HisKA family but the structure of the complex between a mutant version (H188E) of the HK DesK, a member of the minority HisKA_3 family, with the REC domain of DesR[9] has been proposed as the conformational state of this reaction. However, the absence of a phosphoryl group or its mimetic in the structure hinders a detailed deduction on the phosphotransfer mechanism. Finally, signaling is shut down upon RR dephosphorylation mediated by auto-hydrolysis or driven by the HK in the phosphatase reaction[6]. Structural snapshots for the phosphatase reactions have been obtained for the HK–RR complexes HK853–RR468 (PDB: 3DGE[10]) and DesK-DesR (PDB: 5IUN[9]) representatives of the HisKA and HisKA_3 families, respectively. According to the structures, the HK–RR interacting interface in both phosphotransfer and phosphatase reactions is almost conserved but in the phosphotransfer reaction the HK adopts an asymmetric dimeric conformation similar to the one observed in the kinase reaction[2,9], while in the phosphatase reaction the HK dimer is symmetric[2,10]. In line with the structural differences observed among reactions, it has been shown that the phosphotransfer and phosphatase reactions are not reversal as mutations in the phosphorylatable His of HisKA HKs abolishes phosphotransfer but not phosphatase reaction[11]. While in the phosphotransfer reaction the phosphoryl group moves directly from the HK His to the RR Asp, it is basically accepted that the HK-mediated RR dephosphorylation involves the participation of a water molecule, which is activated by a polar residue in the HK, to attack the phosphoryl group bound to the Asp in the RR[3,9,11]. However, there seems to be molecular discrepancies between HK families in this reaction. In HisKA family a Thr/Asn residue in the DHp motif HD/E-X-X-T/N works as the activating polar residue while the phosphorylatable His works enhancing the phosphatase activity by positioning the nucleophilic water as was suggested by the structure of HK853–RR468 complex[10,11]. Meanwhile in HisKA_3 family, a Gln located in the HD-X-X-X-Q/H motif is crucial for the reaction but the phosphorylatable His does not play any role as it is further away from the active site as observed in the structure of DesK–DesR complex[9]. In this scenario, it is intriguing how the signal exquisitely controls the balance between three different reactions, even more when phosphotransfer and phosphatase are mechanistically distinct reactions.

The versatility on the reactions catalyzed by TCS is facilitated by the chemistry of the phosphorylatable His/Asp residues[12,13]. Unlike eukaryotic signaling, which is dominated by Ser/Thr/Tyr phosphorylation that generates highly stable phosphoester bonds (P–O), phosphorylation in His generates a phosphoramidate bond (P–N) less thermodynamically stable and probably more labile due to the capacity of the imidazole nitrogen to become protonated at physiological pHs (acidic ionization constant around $pK_a \sim 6.0$), making the histidine a good leaving group[12] (Fig. 1a). In this way, the P–N shows higher propensity to transfer the phosphoryl group than P–O, speeding up signaling processes. Thus, the level of ionization for the His in the enzymatic reactions is relevant as in the protonated form, the His can act as an acid, donating protons, while in the neutral form can act either as a base, assisting in the removal of protons, or as a nucleophile[14]. However, the mechanism by which the signal could exploit the chemistry of the phosphorylatable residues and, consequently, modulate the signaling by TCSs is poorly understood.

Recently, a new molecular mechanism that gates the phosphatase activity of HKs from HisKA family has been proposed[15]. This mechanism was based on the structure of the complex between HK853 and RR468 bound to the phosphomimetic $BeF_3^-$, solved at pH 5.0, where the authors observed that the side chain of the phosphorylatable His adopted a *gauche−* χ1 rotamer (hereafter, χ1 will be omitted for simplicity) pointing away to the phosphoacceptor Asp. The authors pointed out that the acquisition of this side-chain conformation for the phosphorylatable His was induced by the decrease of the pH, since a previous structure of HK853–RR468, solved at pH 5.6, presented a *trans* χ1 rotamer (hereafter, χ1 will be omitted for simplicity). Then, they correlated the His side-chain conformation in a *gauche−* rotamer at low pH with a decreased in the phosphatase activity proposing a model named pH-gated conformational switch for this reaction. As loss of the phosphatase activity at pH 5.0 was also observed for EnvZ towards OmpR, thus, the authors suggested that the pH-gated conformational switch was conserved in the HisKA family. Overall, in the pH-gated model the authors proposed that the transition between inactive to active states for the phosphatase activity was driven by a mere change on pH, more precisely between 5.2 and 6.5, to produce a turn in the side-chain conformation of the phosphorylatable His from a *gauche−* to a *trans* rotamer (Fig. 1b). Thus, we have explored the relation between the effect of the pH and His rotamer-dependent phosphorylation from a functional and structural perspective solving the structure of several HK853–RR468 complexes at different pHs. In parallel, we have evaluated the effect of the pH in the kinase, phosphotransfer and phosphatase activities for HK853 and EnvZ. So far, our results do not show any correlation between pH and His rotamer disposition since the disposition of this residue is not influenced by the environmental pH in our structures. Moreover, analysis of HKs structures of HisKA family, already deposited in the PDB, confirms that His rotamer disposition is not influenced by the pH, ruling out a pH-gated model. Oppositely, our structural data shows that the *gauche−* conformation for the His can be also adopted after phosphorylation as an inactive resting state. Finally, our biochemical results favor the view that the effects of the pH are due to the ability for the His to act as a general base or as a nucleophile in the reactions catalyzed by the HK in the signaling process.

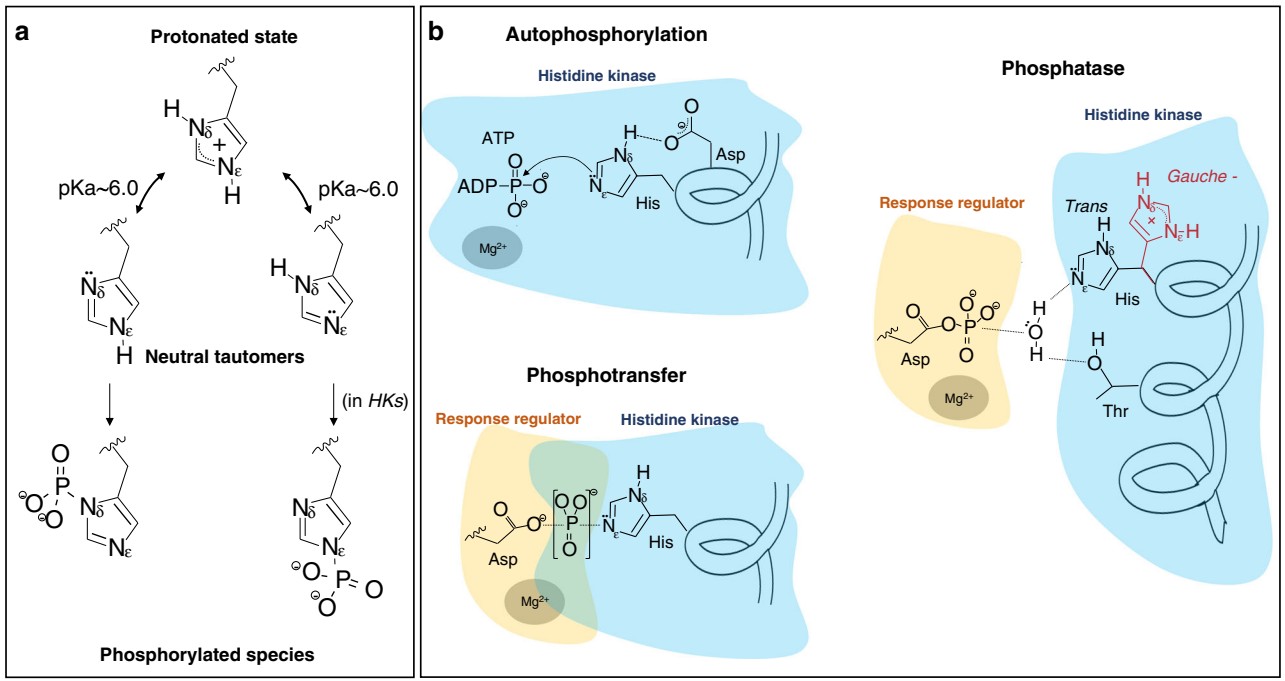

**Fig. 1 Chemical versatility of the His side-chain and its role in HKs activities. a** Schematic representation of protonation and tautomerization states as well as phosphorylated species of His. Structural and functional studies on HKs support autophosphorylation at Nε[7,8,50] while autophosphorylation at Nδ happens at other enzymes such as nucleoside diphosphate kinases[51]. **b** Schematic representation of the autophosphorylation, phosphotransfer, and phosphatase reaction mechanism of HisKA HKs. In the autophosphorylation, the nucleophilic character of His is enhanced by interactions with an adjacent acidic residue (Asp, herein) to attack γ-P ATP; the phosphotransfer is represented as a transition state between the catalytic residue Asp and His with phosphoryl group; in the phosphatase, a water molecule attacks the catalytic Asp of RRs thanks to its activation by a conserved polar residue (Thr) and the catalytic His. Two distinct rotamers *trans* and *gauche*− are shown for the catalytic His, where just the *trans* rotamer is involved in the reaction.

## Results

**Rotamer disposition of phosphorylatable His in HisKA HKs.**
The pH-gated model proposes that in HKs of the HisKA family the rotamer disposition of the phosphorylatable His is regulated by pH, acquiring, an inactive *gauche*− rotamer at pH between 5.2 and 6.5 and an active *trans* rotamer at pHs above 6.5[15] for the phosphatase activity (Fig. 1b). To evaluate this proposition, we analyzed the rotamer disposition of the phosphorylatable His from the 28 structures of HisKA family HKs deposited at the PDB, which comprise EnvZ and CpxA from *Escherichia coli*, HK853 from *Thermotoga maritima*, WalK from *Lactobacillus plantarum* and VicK (WalK) from *Streptococcus mutans*, RetS from *Pseudomonas aeruginosa* as well as DHp domains of PhoR and ERS1 from *Mycobacterium tuberculosis* and *Arabidopsis thaliana*, respectively. As it can be observed in Table 1, the phosphorylatable His in these structures adopt either *gauche*− (27 His) or *trans* (39 His) rotamers. Mostly, the His at both subunits in the dimeric HK presents identical rotamer (15 structures). However, in 10 structures, which include representative of HKs HK853, VicK, CpxA, and ERS1, the His in each subunit of the dimer presents distinct rotamer, supporting that pH is not controlling the side-chain conformation of the His. Moreover, the His in two structures (PDBs; 4JAV[16] for HK853 and 5C93[17] for WalK) shows double *gauche*− and *trans* conformation, indicating that some molecules in the crystal presented either one or the other rotamer. Finally, RetS crystal structure (PDB 6DK7) presents four HK dimers in the asymmetric unit, showing His *gauche*− rotamer four subunits and *trans* the other four. Inspection of the crystallization conditions for all the structures revealed that pH ranged from 4.6 to 8.5 and that the side chain of the phosphorylatable His fluctuates between *gauche*− and *trans* rotamers with no evident correlation with the pH (Table 1). Finally, contrary to the proposal of the pH-gated model, the

structures crystallized at the lowest pH, 4 and 4.6, corresponding to a chimeric EnvZ[18] (containing HAMP domain of Af1503 and the DHp-CA domains of EnvZ) and the isolated EnvZ HAMP-DHp domain[19], respectively, presents *trans* rotamers for the phosphorylatable His, while the *gauche*− rotamer is present in two structures of CpxA crystallized at higher pH (8.5)[8] (Table 1). In those cases where the phosphorylatable His is trapped performing the autophosphorylation conformation, as in subunit A of CpxA (PDB: 4BIW[8], solved at pH 8.5) and subunit A of a chimeric EnvZ (residues L254 to Y265 interchanged by HK853 residues A271–E290; PDB: 4KP4[7], solved at pH 7.5), or the phosphatase competent reaction in HK853 (PDB: 3DGE[10], solved at pH 5.6), it shows an invariably *trans* rotamer, that would account for an active state of the His to perform either autophosphorylation or phosphatase activity. All together, the analysis of the available structures of HisKA family HKs suggests that the acquisition of the two distinct His rotamers seems not be related with the pH but might be related with the catalytic state trapped in the crystal ruling out a pH-gated model.

**HK853–RR468 structures show a pH independent His disposition.** Our previous analysis of the reported HisKA structures does no correlate pH with disposition of His rotamers. However, it might be possible that the pH-gated model was not general but restricted to a subset of HKs. The proposed pH-gated model[15] was structurally supported by discrepancies between the *gauche*− rotamer for the phosphorylatable His (His260) observed in the HK853 structure in complex with RR468 solved at pH 5.0 (HK853–RR468[5.0]; PDB:5UHT) and the *trans* rotamer observed for His260 in the structure solved at pH 5.6 (HK853–RR468[5.6]; PDB: 3DGE)[15]. In the HK853–RR468[5.0] the *gauche*− rotamer was not competent for the phosphatase reaction in contrast to the

**Table 1 His rotamer disposition for HisKA HKs deposited at the PDB.**

| PDB | Protein | pH | His rotamer for each subunit in dimer | Crystallization mother liquor |
|---|---|---|---|---|
| 3DGE | HK853 | 5.6 | *trans/trans* | 1.7 M $(NH_4)_2SO_4$, 2.5% dioxane, 0.1 M citrate, pH 5.6[41] |
| 5UHT | HK853 | 5 | *gauche−/gauche−* | 0.1 M citric acid (pH 4), 0.8 M $NH_4SO_4$. Adjusted final pH to 5.0[15] |
| 2C2A | HK853 | 6.5 | *gauche−/ gauche−* | 1.25 M $LI_2SO_4$, 0.1 M $NH_4$ acetate pH 6.5[26] |
| 4JAU | HK853 | 8.5 | *gauche−/trans* | 8% PEG4000, 0,8 M $LiCl_2$, 0.1 M Tris–HCl pH 8.5[16] |
| 4JAS | HK853 | 5.5 | *gauche−/ gauche−* | 2,2 M $(NH_4)_2SO_4$, 0.1 M Bis-Tris pH 5.5[16] |
| 4JAV | HK853 | 5.5 | *gauche−/gauche−* & *trans* | 2,2 M $(NH_4)_2SO_4$, 0.1 M Bis-Tris pH 5.5[16] |
| 4I5S | VicK | 7.6–8.6 | *trans/gauche−* | 2.3–2.9 M Na formate, 3% PEG 4000[20] |
| 4U7N | WalK | 5.6 | *gauche−/trans* | 1.0 M $(NH_4)_2SO_4$, 1% PEG 4000, 50 mM Bis-Tris pH 5.6[17] |
| 5C93 | WalK | 5.6 | *gauche−* & *trans/trans* | 1.0 M $(NH_4)_2SO_4$, 1% PEG 4000, 50 mM Bis-Tris pH 5.6[17] |
| 4U7O | WalK | 5.6 | *trans/trans* | 1.0 M $(NH_4)_2SO_4$, 1% PEG 4000, 50 mM Bis-Tris pH 5.6[17] |
| 4ZKI | WalK | 5.6 | *trans/gauche-* | 1.0 M $(NH_4)_2SO_4$, 1% PEG 4000, 50 mM Bis-Tris pH 5.6[17] |
| 3ZRV | EnvZ[HAMP-DHp] | 7 | *trans/trans* | 20% PEG 4000, 20% Isopropanol, 0.1 M tri-sodium citrate pH 5.6[19] |
| 3ZRX | EnvZ[HAMP-DHp] | 4.6 | *trans/trans* | 30% MPD, 0.02 M $CaCl_2$, 0.1 M sodium acetate[19] |
| 3ZRW | EnvZ[HAMP-DHp] | 7 | *trans/trans* | 0.4 M Mg formate, 0.1 M Bis-Tris, pH 7[19] |
| 5B1N | EnvZ[DHp] | 6.9 | *trans/trans* | 1.1–1.3 M Na/K phosphate[52] |
| 5B1O | EnvZ[DHp] | 6.9 | *trans/trans* | 0.1 M magnesium formate, 15% PEG 3350[52] |
| 4KP4 | EnvZ[chim] | 7.5 | *trans/trans* | 1.5 M $(NH_4)_2SO_4$, 2% PEG 1000, 2% PEG 4000, 0.03 M Na acetate, 0.1 M HEPES pH 7.5[7] |
| 4BIU | CpxA | 8.5 | *trans/trans* | 1.75 M $(NH_4)_2SO_4$, 25% glycerol, 0.1 M Tris–HCl pH 8.5[8] |
| 4BIW | CpxA | 8.5 | *trans/gauche−* | 1.75 M $(NH_4)_2SO_4$, 25% glycerol, 0.1 M Tris–HCl pH 8.5[8] |
| 5LFK | CpxA | 8,5 | *trans/trans* | 1.5 M $(NH_4)_2SO_4$, 12 % glycerol, 0.1 M Tris–HCl pH 8.5[53] |
| 4BIV | CpxA | 8.5 | *trans/gauche−* | 1.75 M $(NH_4)_2SO_4$, 25% glycerol, 0.1 M Tris–HCl pH 8.5[8] |
| 4CB0 | CpxA | 8.5 | *trans/gauche−* | 1.75 M $(NH4)_2SO_4$, 25% glycerol, 0.1 M Tris–HCl pH 8.5[8] |
| 4BIX | CpxA | 8.5 | *gauche−/gauche−* | 25% PEG3350, 0.2 M $Li_2SO_4$, 0.1 M Tris–HCl pH 8.5[8] |
| 4BIY | CpxA | 8.5 | *gauche−/gauche−* | 25% PEG3350, 0.2 M $Li_2SO_4$, 0.1 M Tris–HCl pH 8.5[8] |
| 4CTI | EnvZ (HAMP[Af1503]) | 4.0 | *trans/trans/trans/trans* | 20% PEG 3350, 0.2 M lithium acetate, 0,1 M MMT buffer pH 4.0[18] |
| 5UKV | PhoR[DHp] | 7.2 | *gauche−/gauche−* | 35% PEG 200, 2 mM EDTA, 0.2 M KI, 0.1 M Na/K phosphate pH 7.2[21] |
| 4MT8 | ERS1[DHp] | 7.5 | *gauche−/trans* | 9% PEG 3350, 0.18 M l-proline, 0.1 M HEPES pH 7.5[54] |
| 6DK7 | RetS [DHp-CA] | 7.5 | *trans/trans/trans/trans/gauche−/ gauche−/gauche−/gauche−* | 2.7 M NaCl, 9 mM $CoCl_2$, 90 mM HEPES pH 7.5[55] |

HK853–RR468[5.6] structure where the *trans* rotamer would seem competent for this reaction. Thus, the authors correlated the pH decrease below 5.5 with the acquisition of the phosphatase inactive *gauche−* rotamer. To confirm if the pH regulates the His260 side-chain conformation in HK853 and, consequently, the pH-gated model for this HK, we solved the crystal structure of the complex HK853–RR468 at pHs 5.5, 6.5, 7.0 and 7.5 (herein HK–RR[pHs] structures) in the presence of ADP and $BeF_3^−$ as in HK853–RR468[5.0]. Crystals at each pH diffracted X-rays at good resolution (~2.0 to 2.85 Å) and all of them contained, in the asymmetric unit, a symmetric dimer of HK853 bound to two molecules of RR468 (Fig. 2a and Table 2). The overall structure of the complex at each pH was similar to the previous structure of HK853−RR468[5.6] and HK853−RR468[5.0] complexes, respectively (Fig. 2b, c and Supplementary Table 1) and as expected, ADP and $BeF_3^−$ were present at the active centers of HK853 and RR468, respectively (Fig. 2a, d, f). A close inspection of His260 in the HK–RR[pHs] structures showed the presence of the inactive *gauche− rotamer*, as observed in the low pH HK853−RR468[5.0] structure, independently of the pH used for crystallization (Fig. 2a, c) different from the *trans* rotamer observed in the HK853−RR468[5.6] structure (Fig. 2b, e). This observation ruled out a pH-induced mechanism in HK853 to acquire a *trans* or *gauche−* rotamer for the phosphorylatable His.

According to the pH-gated model, the relative disposition of the rotamer in the phosphorylatable His was associated with slight movements at the N-terminal helix α1 of HK853 that affected DHp packing and caused rotation of the CA domain[15]. These HK movements resulted in a distance shift (~2.7 Å) of RR468 for the phosphatase-non-competent structure, proposing a molecular explanation that would correlate low pH with the lack of phosphatase activity[15]. To check the correlation between pH and the relative HK–RR disposition we superposed the HK–RR[pHs] with the phosphatase-non-competent HK853–RR468[5.0] and the phosphatase-competent HK853–RR468[5.6] structures (Fig. 2b, c, Supplementary Table 1). This comparison revealed that all the structures, independently of the pH, showed higher similarity with the phosphatase-competent HK853–RR468[5.6] than with HK853–RR468[5.0] (RMSD values ~0.7 vs. ~2.0 Å for the superposition of ~707 residues, respectively). In the HK–RR[pHs] structures the DHp helix α1 and CA domains in HK853 adopt an identical conformation to HK853–RR468[5.6] and the two molecules of RR468 are also placed close to the DHp domain as in HK853–RR468[5.6] structure with no inter-domain rearrangements (Fig. 2b). Thus, slight differences in the relative HK–RR disposition were just observed between HK–RR[pHs] and HK853–RR468[5.0] as has been described previously[15] (Fig. 2c). Since the HK–RR[pHs] structures present a *gauche−* rotamer for His260 as in HK853–RR468[5.0], it is clear that the acquisition of *gauche−* or *trans* rotamers do not correlate with the environmental pH in HK853 or the slight conformational changes observed between HK853–RR468[5.0] and HK853–RR468[5.6] structures and, consequently, cannot provide molecular support for the pH-gated model.

**Phospho-His can be stabilized in the *gauche−* rotamer.** If pH does not regulate the side-chain conformation between *trans* and *gauche−* rotamers for the phosphorylatable His, then, what determines the rotamer disposition? In order to answer this question we analyzed the His side chain interactions. HK853 structures reported here and the previously deposited in the PDB show that His260 side chain interact with the main chain oxygen of residue at position −4 (A256 in HK853) when adopts

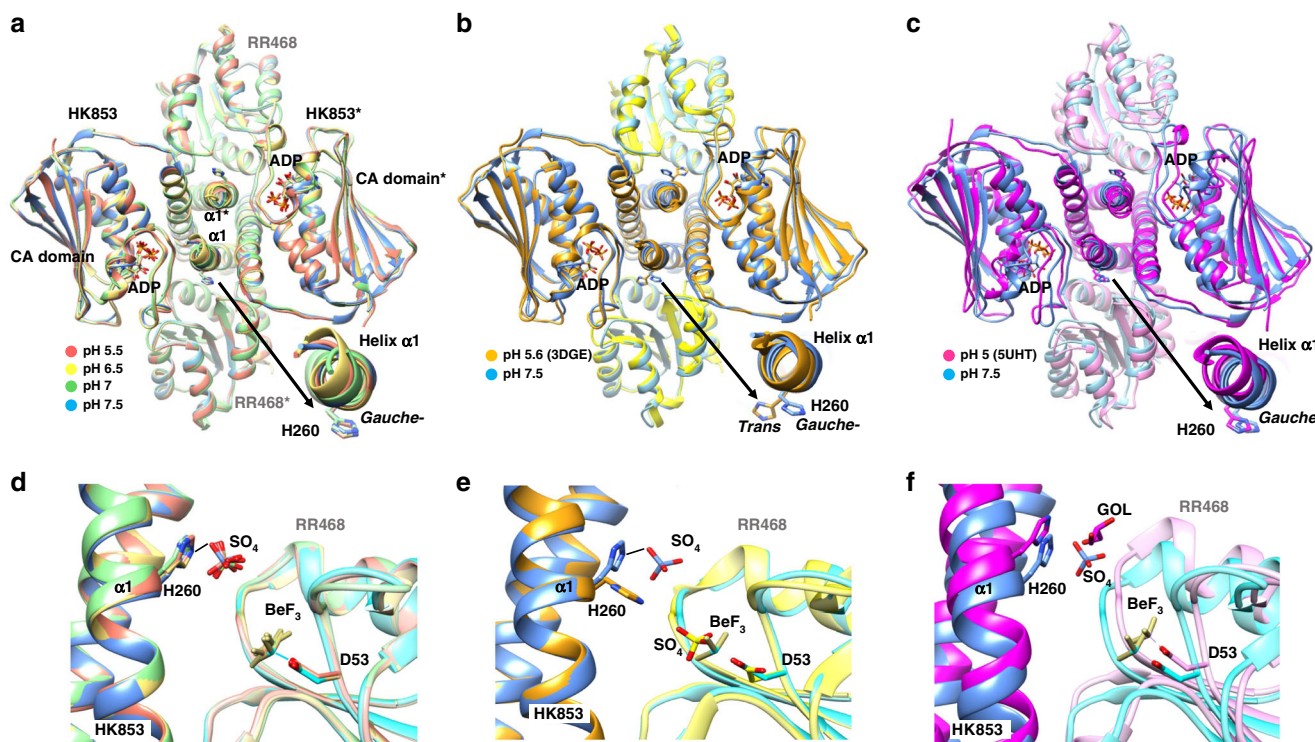

**Fig. 2 Structures of the wild-type complex of HK853–RR468 at different pHs. a** Superposition of HK–RR[pHs] structures. The complexes composed of a dimeric HK853 bound to two molecules of RR468 (the asterisk denotes the second molecule) solved at pHs 5.5 (in salmon), 6.5 (yellow), 7 (green), and 7.5 (blue) are represented in cartoon with HKs and RRs in dark and pale hues, respectively. The HK phosphorylatable His (H260) in the helix α1 of DHp domain and the ADP in the CA domain are shown in sticks colored by atom type with the carbons in the same color as the corresponding molecule. A close view of the H260 for one monomer is shown. **b** Superposition of HK–RR[pH] at pH 7.5 (blue tones) with HK853–RR468[5.6] (orange-yellow tones) highlighting the *gauche−* and *trans* rotamers of the H260, respectively. **c** Superposition of HK–RR[pH] at pH 7.5 (blue tones) with HK853–RR468[5.0] (magenta tones) highlighting the *gauche−* rotamer for H260. **d** Close view of the active centers for the superposition of HK–RR[pHs] structures. The sulfate ion (labeled as SO₄; the two negative charges of the anion have been omitted for clarity) interacting with the HK H260 and the phosphomimetic BeF₃⁻ bound to the phosphoacceptor D53 of RR468 are shown in sticks. Structures are colored by pHs following the same code as in **a**. **e** and **f** Close view of the active centers for the superposition of HK–RR[pHs] at pH 7.5 (blue tones) with HK853–RR468[5.6] (orange-yellow tones) in **e** and with HK853–RR468[5.0] (magenta tones) in **f**. The polar molecule glycerol (GOL) located in HK853–RR468[5.0] at similar position as sulfate ion bound to H260 in HK–RR[pHs] is shown in sticks with carbons in the same color as the corresponding molecule. The phosphomimetic BeF₃⁻ (labeled as BeF₃) bound to RR468 D53 is also shown in sticks.

the *gauche−* rotamer. Similar interactions are observed for the phosphorylatable His in the CpxA (PDB 4BIX)[8], WalK (PDB 5C93)[17], VicK (PDB 4I5S)[20], and PhoR (PDB 5UKV)[21] HKs, supporting that this interaction would be stabilizing the rotamer disposition. Close inspection of the His260 in the HK–RR[pHs] structures showed that this residue was also interacting with a sulfate ion through its side-chain Nε (Fig. 2d). The sulfate ion was present at high concentration in the crystallization conditions (ammonium sulfate ≤ 1.8 M) and, as it is known, sulfate ions tend to occupy the position of phosphates in crystal structures mimicking its actions in several biological processes such as protein phosphorylation. In this way, sulfate ions have been found occupying the position of the phosphoryl group in phospho-Ser/Thr[22,23], phospho-His[24] or phospho-Asp[10,25] among others. In the HK–RR[pHs] structures, the sulfate ion interacting with H260 was stabilized by a hydrogen bond network which involved contacts with the main chain nitrogen of G86 and the side chain of D90, both at loop β4α4 of RR468, as well as with several water molecules (Supplementary Fig. 1a). Sulfate ions were also found in identical positions at the structures of isolated HK853 (PDB 2C2A)[26] and a rewired HK853–RR468 complex (PDB 4JAV)[16] solved at pH 6.5 and 5.5, respectively. These sulfate ions interacted with the His260 having a *gauche−* rotamer, although in the rewired complex His260 presented both *trans* and *gauche−* rotamers, but in both structures the sulfate ion

was further stabilized through additional salt-bridges with R314 and R317 (Supplementary Fig. 1b). Remarkably, in the HK853–RR468[5.0] structure, a glycerol molecule, which also mediated interaction with His260, was found at a similar position than the sulfate (Fig. 2f), indicating that this site presents a correct environment to accommodate polar molecules. In order to confirm that the sulfate ion was indeed stabilized in that position thanks to the presence of the phosphorylatable His in a *gauche−* rotamer, we generated mutants of HK853 and RR468 defective in its phosphorylatable residues H260A and D53A, respectively, and obtained the structures of its complex (HK853[H260A]–RR468[D53A]) at pH 5.3 and 7.5 (Table 3). The overall structures of the complexes were similar to the HK–RR[pHs] (Fig. 3a) but the absence of the imidazole group due to mutation of His to Ala was concomitant with absence of sulfate ion bound, supporting that the presence of the His was required to coordinate the sulfate (Fig. 3b). A second sulfate ion bound to the catalytic site of RR468 was also observed, despite absence of phosphorylatable Asp due to its mutation to Ala (Fig. 3b). The presence of sulfate ions at the active center of RRs lacking the phosphorylatable Asp has been previously reported for MaeR triggering its phosphorylated conformation[25]. Indeed, comparing the structure of RR468[D53A] with RR468 bound to BeF₃⁻ (PDB 3GL9)[10] confirmed that the former presented the phosphorylated conformation (Supplementary Fig. 2). Additionally, we confirmed that the absence of the sulfate

**Table 2 Data collection and refinement statistics for HK–RR$^{pHs}$ structures.**

|  | HK853-RR468 pH 7.5 | HK853-RR468 pH 7 | HK853-RR468 pH 6.5 | HK853-RR468 pH 5.5 |
|---|---|---|---|---|
| *Data collection* |  |  |  |  |
| Space group | I 2 | I 2 | I 2 | I 2 |
| Cell dimensions |  |  |  |  |
| $a, b, c$ (Å) | 68.47, 92.71, 174.52 | 68.67, 93.57, 172.70 | 68.74, 93.61, 173.93 | 68.62, 92.93, 174.34 |
| $\alpha, \beta, \gamma$ (°) | 90, 93.39, 90 | 90, 93.32, 90 | 90, 93.35, 90 | 90, 93.49, 90 |
| Resolution (Å) | 87.11–2.2 (2.27-2.2) | 30.0–2.83 (2.92-2.83) | 28.9–2.35 (2.43-2.34) | 49.21–2.87 (2.97-2.87) |
| Total no. of reflections | 208,498(17,468) | 71,591(9381) | 208,226(16,456) | 92,410(14,022) |
| $R_{merge}$ | 0.055 (0.539) | 0.119 (0.451) | 0.063 (0.528) | 0.064 (0.280) |
| $R_{meas}$ | 0.074 (0.729) | 0.152 (0.567) | 0.073 (0.648) | 0.079 (0.343) |
| $I/\sigma I$ | 11.6 (2.1) | 6.0 (2.3) | 13.7 (2.6) | 14.1 (5.2) |
| $CC_{1/2}$ | 0.999 (0.837) | 0.986 (0.709) | 0.999 (0.839) | 0.998 (0.943) |
| Completeness (%) | 99.8 (99.8) | 95.0 (86.1) | 99.2 (96.7) | 97.6 (98.5) |
| Redundancy | 3.8 (3.9) | 2.9 (2.9) | 4.6 (3.8) | 3.8 (3.9) |
| *Refinement* |  |  |  |  |
| $R_{work}/R_{free}$ | 0.186/0.227 | 0.243/0.309 | 0.215/0.265 | 0.240/0.287 |
| No. of atoms | 5997 | 5801 | 5886 | 5832 |
| Protein | 5546 | 5495 | 5534 | 5535 |
| Ligand/ion | 142 | 134 | 135 | 124 |
| Water | 309 | 172 | 217 | 173 |
| Average $B$-factors (Å$^2$) | 59.2 | 67.9 | 66.1 | 65.5 |
| Protein | 59.2 | 68.5 | 66.4 | 66.1 |
| Ligand/ion | 66.8 | 71.2 | 68.5 | 65.1 |
| Water | 55.0 | 44.2 | 58.6 | 44.6 |
| R.m.s. deviations |  |  |  |  |
| Bond lengths (Å) | 0.02 | 0.012 | 0.008 | 0.012 |
| Bond angles (°) | 1.97 | 1.61 | 1.23 | 1.12 |
| PDB code | 6RGY | 6RFV | 6RGZ | 6RH0 |

**Table 3 Data collection and refinement statistics for HK853$^{H260A}$ and RR468$^{D53A}$ complexes.**

|  | HK853-RR468$^{D53A}$ pH 7 | HK853-R468$^{D53A}$ pH 5.3 | HK853$^{H260A}$–RR468$^{D53A}$ pH 7.5 | HK853$^{H260A}$–RR468$^{D53A}$ pH 5.3 |
|---|---|---|---|---|
| *Data collection* |  |  |  |  |
| Space group | I 2 | I 2 | I 2 | I 2 |
| Cell dimensions |  |  |  |  |
| $a, b, c$ (Å) | 68.37, 91.15, 176.15 | 68.53, 92.13, 175.93 | 68.54, 91.47, 175.37 | 68.55, 92.19, 175.02 |
| $\alpha, \beta, \gamma$ (°) | 90, 93.6, 90 | 90, 93.47, 90 | 90, 93.56, 90 | 90, 93.61, 90 |
| Resolution (Å) | 87.9–2.00 (2.04-2.00) | 87.8–2.00 (2.04-2.00) | 87.51–2.0 (2.04-2.00) | 87.33–1.9 (1.93-1.90) |
| Total no. of reflections | 409,524(26,968) | 272,904(17,175) | 407,319(25,528) | 256,875(27,358) |
| $R_{merge}$ | 0.085 (1.156) | 0.053 (0.460) | 0.077 (0.771) | 0.061 (0.727) |
| $R_{meas}$ | 0.097 (1.289) | 0.066 (0.557) | 0.089 (0.884) | 0.070 (0.815) |
| $I/\sigma I$ | 10.4 (1.6) | 13.5 (2.9) | 11.2 (2.4) | 14.4 (2.4) |
| $CC_{1/2}$ | 0.998 (0.718) | 0.998 (0.855) | 0.997 (0.887) | 0.998 (0.822) |
| Completeness (%) | 98.0 (97.1) | 99.5 (99.8) | 99.8 (99.9) | 98.8 (98.0) |
| Redundancy | 5.7 (5.9) | 3.7 (3.8) | 5.6 (5.7) | 6.2 (6.2) |
| *Refinement* |  |  |  |  |
| $R_{work}/R_{free}$ | 0.211/0.255 | 0.192/0.228 | 0.182/0.229 | 0.177/0.2192 |
| No. of atoms | 5976 | 5996 | 6125 | 6369 |
| Protein | 5524 | 5530 | 5626 | 5719 |
| Ligand/ion | 104 | 114 | 109 | 149 |
| Water | 348 | 352 | 390 | 501 |
| Average $B$-factors (Å$^2$) | 51.3 | 42.9 | 49.9 | 47.1 |
| Protein | 50.9 | 42.5 | 49.7 | 46.2 |
| Ligand/ion | 56.1 | 43.5 | 52.5 | 61.1 |
| Water | 56.8 | 49.0 | 52.2 | 52.6 |
| R.m.s. deviations |  |  |  |  |
| Bond lengths (Å) | 0.012 | 0.007 | 0.020 | 0.019 |
| Bond angles (°) | 1.59 | 1.12 | 1.97 | 1.92 |
| PDB code | 6RH1 | 6RH2 | 6RH7 | 6RH8 |

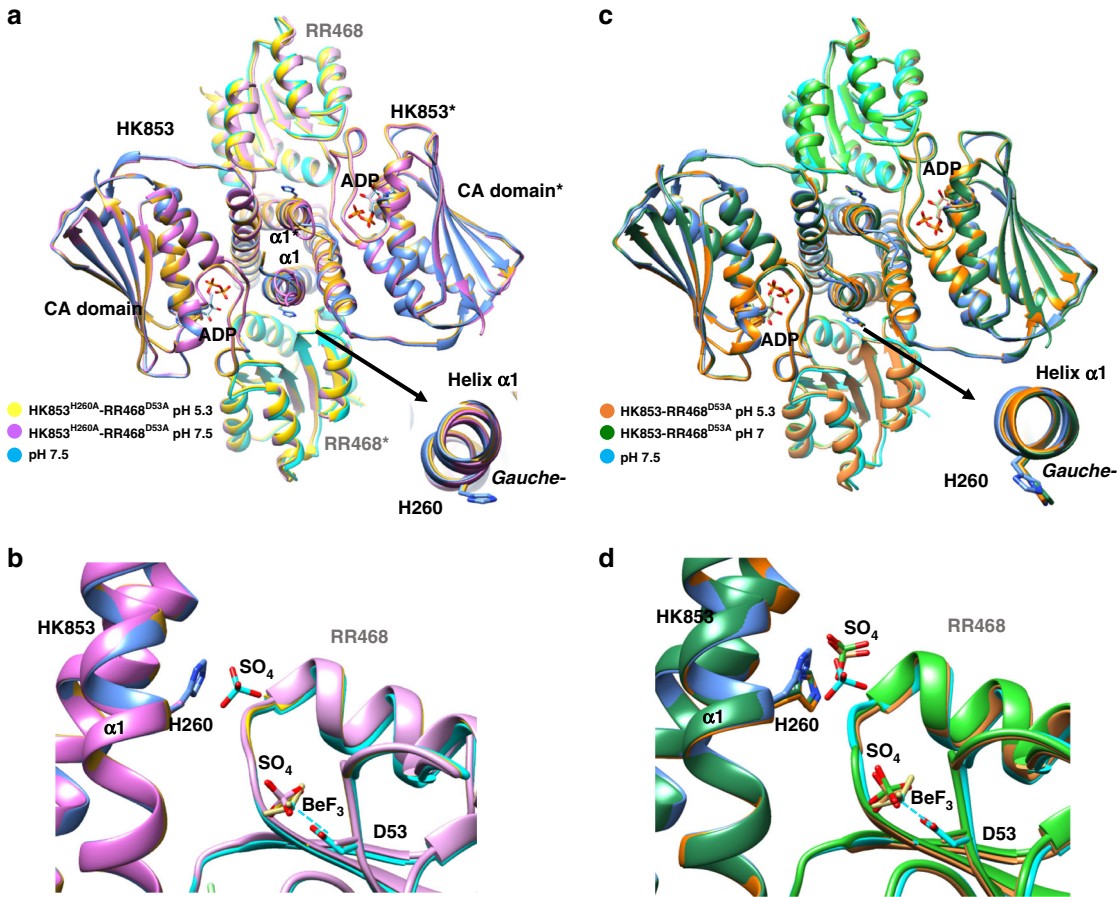

**Fig. 3 Structures of complexes mutated in the phosphorylatable residues at different pHs. a** Superposition of wild-type HK–RR$^{pHs}$ structure at pH 7.5 (blue tones) with HK853$^{H260A}$–RR468$^{D53A}$ at pH 5.3 (yellow tones) and at pH 7.5 (purple tones). Structures are represented in cartoon with HKs and RRs in dark and pale hues, respectively. **b** Close view of the active center of superposed structures in **a** showing absence in the HK853$^{H260A}$–RR468$^{D53A}$ structures of sulfate ion bound to H260 due to mutation to Ala. The sulfate ion in HK–RR$^{pHs}$ structure at pH 7.5 is shown as sticks with sulfur and oxygen atoms in cyan and red, respectively. Oppositely, a sulfate ion (in stick with sulfur atom in magenta and labeled as SO$_4$; the two negative charges of the anion have been omitted for clarity) is found at the active center of RR468 in the HK853$^{H260A}$–RR468$^{D53A}$ structures at similar position than phosphomimetic BeF$_3^-$ (in yellow sticks, labeled as BeF$_3$) despite absence of Asp53 due to mutation to Ala. **c** Superposition of wild-type HK–RR$^{pHs}$ structure at pH 7.5 (blue tones) with HK853–RR468$^{D53A}$ at pH 5.3 (orange tones) and at pH 7 (in blue). **d** Close view of the active center of superposed structures shown in **c** highlighting the presence of the sulfate ion bound to H260 (in stick with sulfur atom colored in similar tone as the corresponding molecule). A second sulfate ion (in stick with sulfur atom colored in similar tone as the corresponding molecule) bound at similar position than phosphomimetic BeF$_3^-$ (in yellow sticks) is found in the RR468 active center of HK853–RR468$^{D53A}$ despite absence of Asp53.

ion in HK853$^{H260A}$ was due to the loss of the His260 imidazole group by solving two additional structures comprising wild type HK853 in complex with RR468$^{D53A}$ (HK853–RR468$^{D53A}$) at pH 5.3 and 7 (Fig. 3c and Table 3). In these two structures, His260 presented a *gauche−* rotamer and was coordinating a sulfate ion in a similar position to that observed in the other complexes (Fig. 3d). Similarly, the sulfate ion was further stabilized by additional contacts with HK853 R314 as well as with RR468 K85 and D90. In these two structures, RR468$^{D53A}$ also showed the presence of the sulfate ion at the catalytic center in an identical position to the one observed in the HK853$^{H260A}$–RR468$^{D53A}$ structures and so had acquired the phosphorylated conformation (Fig. 3d and Supplementary Fig. 2). Finally, we looked for the presence of a sulfate ion coordinated to the phosphorylatable His in other structures of HisKA family and we found that WalK (PDB 5C93), which was crystalized in presence of high concentration of sulfate (1.0 M ammonium sulfate)[17], coordinated a sulfate ion in an equivalent position as HK853. In WalK the phosphorylatable His (His386) presented both *trans* and *gauche−* rotamers but the sulfate ion was coordinated to the *gauche−* rotamer (Supplementary Fig. 3). Overall, these data supports that

the phosphorylated His could be stabilized upon autophosphorylation by adopting the *gauche−* rotamer as an inactive resting state that would be stabilized by the contacts of the phosphoryl group with residues of both HK and RR.

**Effect of pH on the phosphatase activity of HK853–RR468.** According to the pH-gated model, the pH 5.2–6.5 induced a *gauche−* rotamer in the phosphorylatable His that precluded the coordination of the catalytic water and, consequently, hampered the phosphatase activity. To validate biochemically this hypothesis, the authors showed that a HK853 version lacking the CA domain presented phosphatase activity in vitro at pH 8.0 but not at pH 5.0. Our structural data on HK853–RR468 complexes at different pH do not support this model since *gauche−* rotamers are observed independently of the pH, even at neutral pH 7.5 where the pH-gated model favored the *trans* rotamer. To clarify the discrepancies, we re-examined the catalytic activities of the HK853–RR468 TCS system at different pHs. For that purpose, we used a HK853 version with the complete catalytic portion, including CA and DHp domains since it has been shown that the presence of nucleotide, either ADP or ATP, in the CA domain

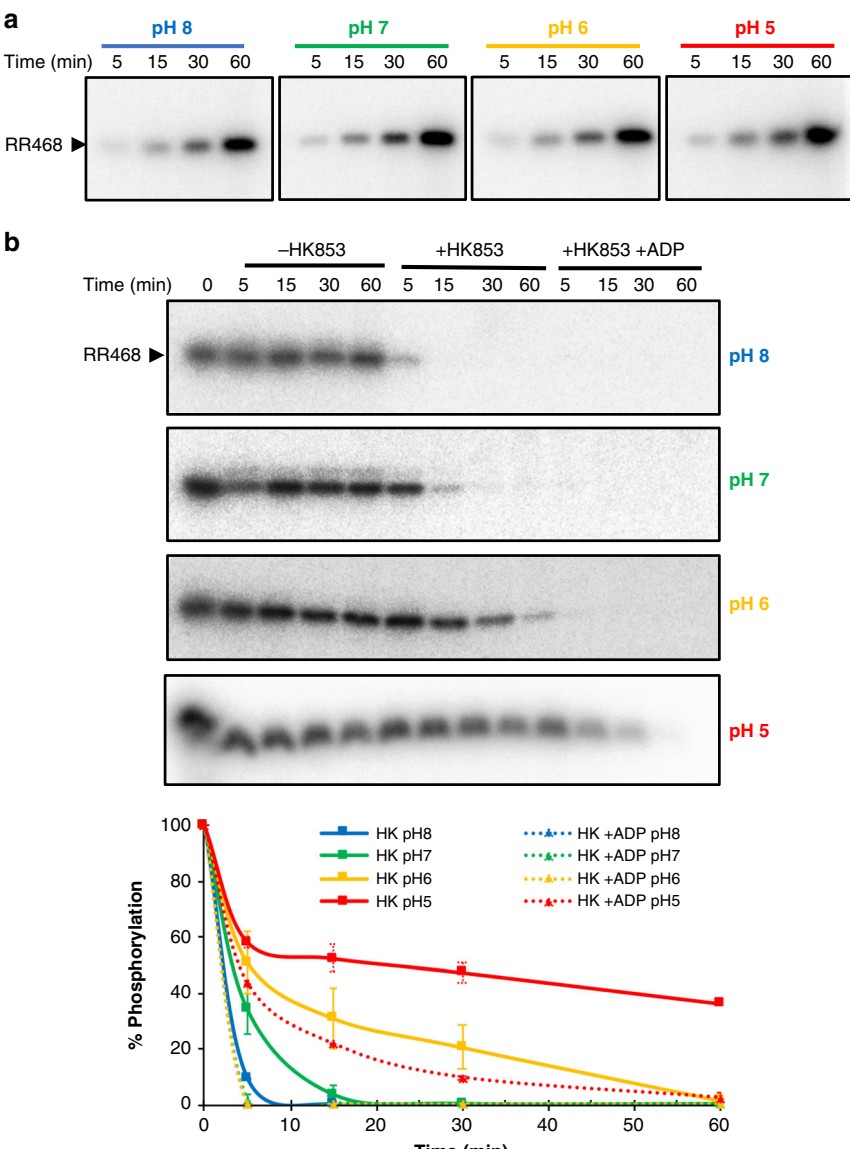

**Fig. 4 RR468 phosphorylation and dephosphorylation at different pHs. a** Time course of RR468 phosphorylation upon incubation with acetyl-phosphate (AcP) as phosphodonor at different pHs. **b** Phosphatase activity of HK853 over phosphorylated RR468 in the absence and presence of ADP. Gels correspond to a representative experiment and quantification of three independent experiments are plotted. Error bars represent SD. Source data are provided as a Source Data file.

works as a cofactor to stimulate the phosphatase activity[27]. First, we checked the autophosphorylation rate of RR468 by the small phosphodonor acetylphosphate (AcP) at different pHs (5–8) using radioactive AcP. Appearance of a band of RR468–P was observed at all pHs which increased over time in a similar trend, reaching a maximum at the end time of the experiment (60 min) and demonstrating the stability of RR468 at the tested pH range (Fig. 4a). The low acidic ionization constant for the phosphoacceptor Asp ($pK_a \sim 3.9$) that allows the nucleophilic attack to the AcP even at pH 5.0 together with the stability of the phosphoester bond P–O at low pH[12] may account for this result. After removing the free radioactive AcP, the HK853 phosphatase activity over RR468–P at different pHs and the effect of the nucleotide was checked. Notice that as it was previously described, the phosphorylated RR was stable at the tested pH range (8–5). Incubation with HK853 at pH 8.0 produced rapid loss of the RR468–P band in only 5 min that was not observed at 15 min (Fig. 4b). At pH 7, incubation with HK853 required additional

time to completely eliminate the RR468–P band. This effect was stronger at lower pHs since at pH 6 required even 30 min to decrease substantially the RR468–P band and at pH 5.0 still conserved ~50% of its intensity after 60 min of incubation (Fig. 4b). However, incubation of RR468–P with HK853 in the presence of 2 mM ADP did enhance the phosphatase activity at all pHs, disappearing the RR468–P band after 5 min of incubation at pH 8, 7 and 6 and latter at pH 5 (Fig. 4b). Thus, addition of ADP clearly stimulated the phosphatase activity, even at pH 5, most probably by the stabilization of the CA domain in a conformation competent to interact with the RR as it was observed in the structures of the HK853–RR468 complexes. Therefore, lowering the pH below 6 increases the His protonation and decreases its capacity to act as a general base, hence, reducing the phosphatase activity. Calculation of the ionization constant $pK_a$ for the imidazole ring in the phosphorylatable His260 on the HK853 structures reported here or available in the PDB showed values mainly comprised between 5.8 and 6.4, close to the

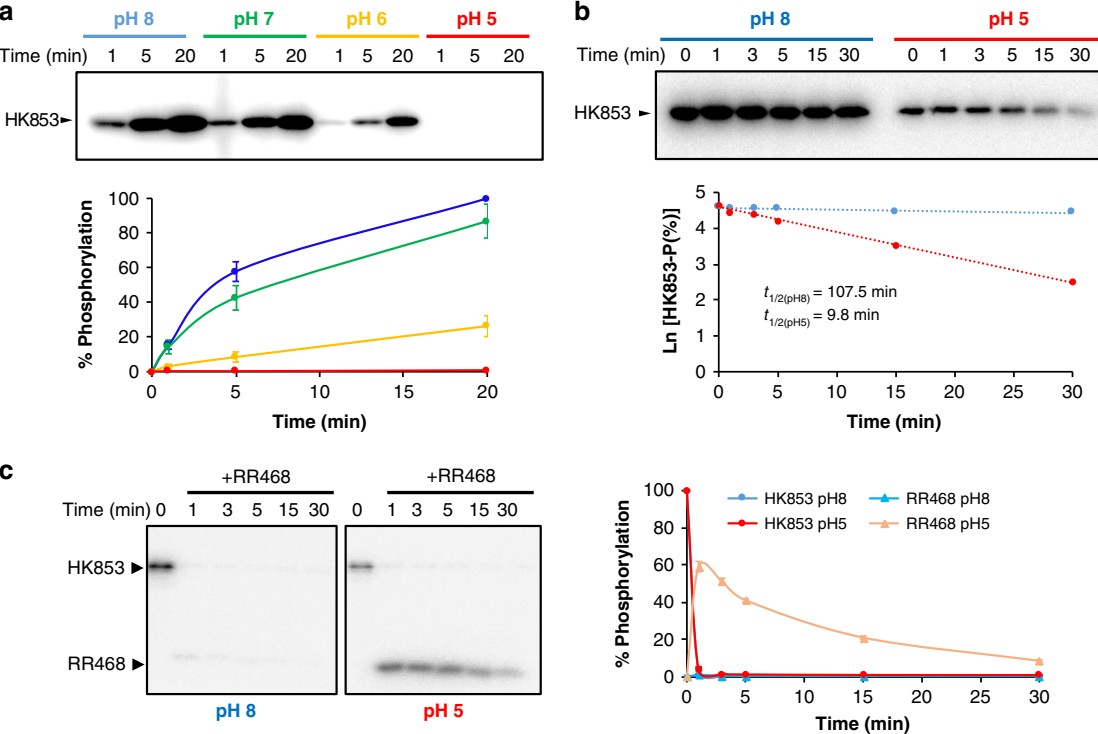

**Fig. 5 Autophosphorylation of HK853 and phosphotransfer to RR468. a** Time course of HK853 autophosphorylation with [γ-$^{32}$P] ATP at different pHs. **b** Stability of phosphorylated HK853 at acidic (5) and basic (8) pH. Phosphorylation was quantified by autoradiography, phosphorylation signals at $t = 0$ were set to 100% for each pH and phosphorylation were calculated accordingly. Values were plotted in a semi logarithmic scale over time, using a linear fit to calculate half-life times of HK853–P at each pH. **c** Time course of phosphotransfer from HK853 to RR468 at acidic (5) and basic (8) pH. Gels correspond to a representative experiment and quantification of three independent experiments are plotted. Error bars represent SD. Source data are provided as a Source Data file.

theoretical p$K_a$ ~ 6.0 for this residue (Supplementary Table 2). Overall, the in vitro assays support that the absence of phosphatase activity for HK853 at pH 5.0 reported in the pH-gated model was due to the absence of CA domain, in the HK853 protein used in the assays, along with the pH effect on the protonation state of the His, but not to a conformational change on this residue.

**pH Influence on HK853 kinase and phosphotransfer activities.** As pH may influence the ionization state of the phosphorylatable His and so the phosphatase activity, we considered checking the effect of pH on the kinase and phosphotransfer activities. The kinase reaction in HKs involves its autophosphorylation by a nucleophilic attack of the His to the γ-phosphate of ATP. In order to work as a nucleophile, the His should release a proton to produce the conjugate base form, that is, to acquire the unprotonated neutral form which is most abundant at a pH higher than 6.0. We checked the HK853 autophosphorylation rate at different pHs from basic to acidic (8, 7, 6, and 5). A clear reduction in autophosphorylation concomitant with the decrease in pH was observable, showing almost null activity at pH 5 (Fig. 5a). To check if the attenuation in HK853 phosphorylation was due to loss of His nucleophilicity that impaired autophosphorylation activity or simply because of the instability of HK853–P at low pH due to the lability of the P–N bond[28], we first phosphorylated HK853 at pH 8.0 and followed the de-phosphorylation rate for another 30 min either at same pH or pH 5.0 (Fig. 5b). Stability of HK853 was not affected by the pH according to circular dichroism spectra (Supplementary Fig. 4), however, at pH 5, the half-life of HK853–P decreased, becoming ~10 min while at pH 8 exceeded hours. Although this shorter HK–P half-life contributed

to the lower phosphorylation observed at acidic pH, it does not fully explain the decrease or absence of autophosphorylation observed concomitant with pH acidification. Therefore, as in the case of the phosphatase reaction, the His protonation prevents this residue to act as a nucleophile, which results in low kinase activity at acidic pHs.

Finally, we explored the influence of pH over the phosphotransfer activity. HK853 was phosphorylated at pH 8, then, buffer was maintained or exchanged to pH 5 and the phosphotransfer reaction was measured after addition of RR468 (Fig. 5c). Upon incubation with RR468, the phosphorylated band for HK853 was lost, either under acidic or basic pH conditions at almost the same rate (>95% transferred at 1 min, the first time point assayed) (Fig. 5c), demonstrating that HK853 could transfer the phosphoryl group to RR468 at both pHs. In this way, and for the time range tested, the phosphotransfer reaction seems to be less dependent of pH at the range assayed (from 8 to 5) than the kinase and phosphatase activity, probably because the hydrolysis of the P–N bond and the phosphoryl transfer is promoted by the attack of the phosphorylatable Asp acidic group which protonation state is weakly affected at pH 5.0 since its p$K_a$ is ~3.9. Moreover, at pH 5 the phospho-His may be a better leaving group than at basic pH given the weakening of the P–N bond due to its propensity to become protonated whereas Asp will still be in its ionized form to catalyze phosphoryl transfer. As mentioned previously, HK853 shows high phosphatase activity towards RR468, thus, upon phosphotransfer the band corresponding to RR468–P is generally absent[16]. In fact, this is the case at pH 8, however, at pH 5, accumulation of RR468–P during the first minutes of reaction could be observed, reflecting the diminished phosphatase activity at low pH (Fig. 5c).

All together our in vitro evaluation of the reactions catalyzed by this TCS showed that lowering the pH to acidic values ~5 have a high impact in autophosphorylation and phosphatase reaction, two reactions that for their catalysis depend mainly on the ionization state of the His. Meanwhile, at the time range tested, it affects only slightly the phosphotransferase activity that is mainly dependent on the Asp residue. Although we could not discard pH dependency for phosphotransferase reaction at shorter times (<1 min), the observations disfavor the control of HK activity by the pH due to a conformational switch as it was proposed in the pH-gated model and supports that the differences simply belong on the chemical nature of the residues involved.

**pH Influence on the functional activities of EnvZ–OmpR.** The pH-gated model was also proposed on the basis of the pH-dependent phosphatase activity of EnvZ–OmpR from *Salmonella typhimurium* where the authors found that EnvZ was a less efficient phosphatase under acidic conditions[15]. Therefore, we decided to evaluate the effect of pH on the functional activities of EnvZ–OmpR from *Escherichia coli*. We first checked the kinase activity of EnvZ at pH 5, 6, 7 and 8. As pH lowers, the autophosphorylation of EnvZ was reduced, showing a reduction of ~50%, ~75%, and almost 90% at pHs 7, 6, and 5, respectively, after 20 min of reaction (Fig. 6a), although it did not completely abolish the autophosphorylation as in the case of HK853 (Fig. 5a). A substantial reduction in EnvZ autophosphorylation induced by lowering the pH (from 7.5 to 5.5) has also been demonstrated previously[29]. Further measurement of the stability for the P–N bond in the phosphorylated EnvZ at pH 5 and 8 showed a similar result as in HK853, where the apparent high-life is reduced from more than 30 to 9 min upon exchange from pH 8 to 5 (Figs. 5b and 6b). As HK853, stability of EnvZ at both pHs was similar according to circular dichroism spectra (Supplementary Fig. 4). Upon autophosphorylation of EnvZ, phosphotransfer activity to

the receiver domain of OmpR was also checked at either pH 5 or 8, showing a similar fast phosphoryl transfer (>85% transferred at 1 min) at both pHs and confirming the less dependent effect of pH for this reaction as in the case of HK853 for the time range tested (1 to 30 min) (Fig. 6c), although we could not dismiss pH dependency at shorter times (<1 min). As EnvZ presents a low phosphatase activity compared to HK853[7], less reduction of OmpR–P bands was observed at either pH, no visible at pH 5 (Fig. 6c). This fact helped us to evaluate the effect of 2 mM ADP on the phosphatase activity of EnvZ. A dramatic reduction of OmpR–P at both pH 8 and 5 was observed confirming the stimulation of the EnvZ phosphatase activity by the nucleotide (Fig. 6c), especially at longer incubation times (Supplementary Fig. 5). However, this activity was enhanced at pH 8 versus 5, as in the case of HK853. Overall, our in vitro functional assays confirm that, as in the case of HK853, the autophosphorylation and phosphatase reactions are impaired at pH 5 while the phosphotransfer activity is less affected at this pH at the time range tested, in close relation with the chemistry of the catalytic residues involved in each reaction but not with a side-chain conformational change of the phosphorylatable His as it is proposed in the pH-gated model.

## Discussion

Histidine is a key amino acid involved in catalytic and regulatory processes, since it is an unique amino acid, its side chain presents a $pK_a$ closest to physiological pH. This particularity, which derives from the chemistry of the imidazole group, explains the presence of this amino acid in the active centers of many proteins playing critical roles in catalysis[30,31], metal binding[32], pH sensing[33], proton conduction[34], or phosphorylation acceptor[6,10]. The imidazole side chain of His can alternate among two tautomeric forms in its neutral anionic state and a fully protonated charged form with changes of few pH units around the physiological pH.

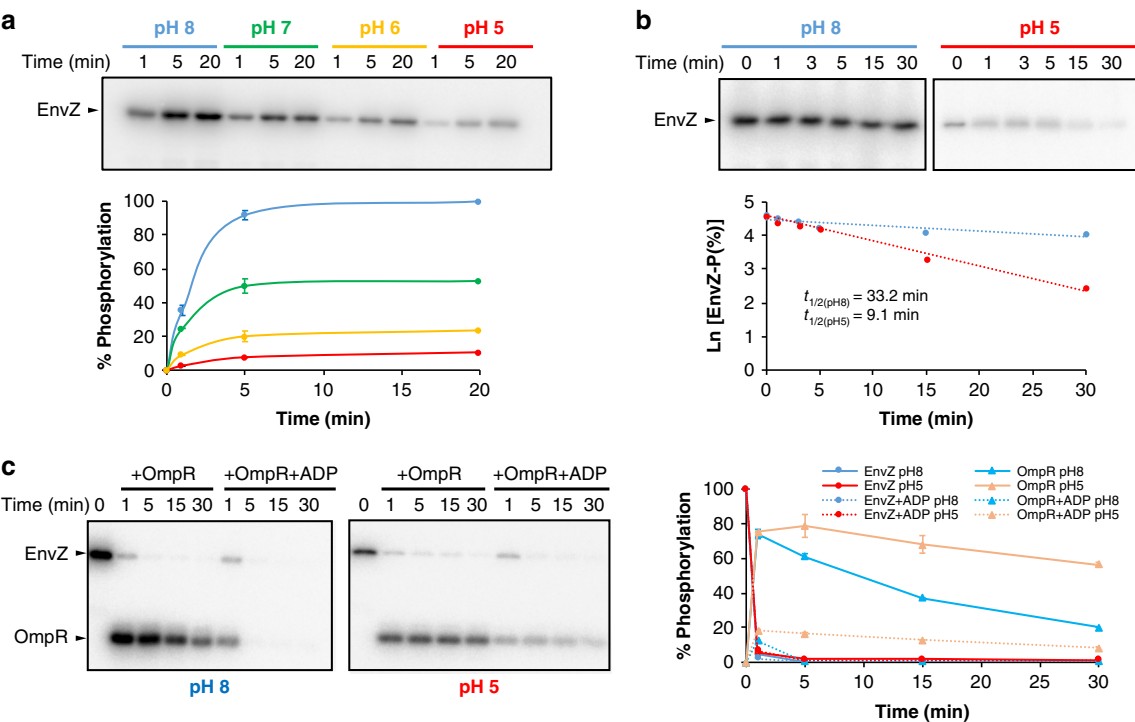

**Fig. 6 Autophosphorylation of EnvZ and phosphotransfer to OmpR. a** Time course of EnvZ autophosphorylation with [γ-$^{32}$P] ATP at different pHs. **b** Stability of phosphorylated EnvZ at acidic (5) and basic (8) pH. Half-life plots for each pH is shown. **c** Phosphotransfer of phosphorylated EnvZ to OmpR at acidic (right) and basic (left) pH in the absence and presence of ADP. Gels correspond to a representative experiment and quantification of three independent experiments are plotted. Error bars represent SD. Source data are provided as a Source Data file.

These conversions together with a combination of ring flips that produce different rotamers, such as *gauche−*, *gauche+*, and *trans*, modulate the His activity[35]. In this sense, the recent proposition of a pH-gated conformational switch to modulate the phosphatase activity of HKs from His_KA family[15] seemed to be a very attractive explanation for the molecular mechanism of this reaction. Indeed, the gating role for His has been described previously, where His switches between rotamers upon ligand binding[36,37] or pH change[38,39] as a result of conversion between protonated-neutral states. However, the structures of HK853–RR468 at varying pHs ranging from 5.3 to 7.5 shows an invariable *gauche−* rotamer for the catalytic His identical to the rotamer observed by Liu and collaborators at pH 5.0, demonstrating that pH is not driving the side-chain conformational switch. Oppositely, the structures support that the *gauche−* rotamer of His260 should represent an inactive resting state for the His awaiting to sense RR binding for phosphotransfer or P-RR for dephosphorylation, albeit independent of pH. In this resting conformation the phospho-His would be stabilized by a network of contacts with the phosphoryl group which are mimicked in our structures by a sulfate ion. Therefore, the reduction in phosphatase activity for HisKA family of HKs observed when pH drops below 6.0 is due to the protonation state of the His, which precludes this residue to activate the catalytic water. Furthermore, if the pH could determine the rotamer disposition of the phosphoacceptor His, the rest of the reactions where this residue participates would be equally altered. This was the case for the autophosphorylation reaction, which is mediated by the nucleophilic attack of the His to the γ-phosphate of ATP, but not for the phosphotransfer reaction that at the same time range is less affected by pH variation between 8 and 5, as the hydrolysis of the P–N bond is promoted by the acidic group of the phosphorylatable Asp which side-chain presents a p$K_a$ much lower than 5. These observations rule out the pH-gated model since if the His conformation was modified by the pH it would be no longer properly aligned with the acceptor Asp, as it was proposed in the pH-gated model for the phosphatase reaction, and consequently, the activity should be modified in the same way for both reactions but this is not the case. Oppositely, this is the case for the autophosphorylation reaction where our data shows that the activity decreases similarly to the phosphatase. This observation is explainable by the chemistry of the residues working as nucleophile or general base in each reaction (His in kinase and phosphatase, and His-Asp in phosphotransferase) but not for the pH-gated model.

Our studies confirm previous observations for the stimulating role of the ATP-binding domain in the phosphatase activity of EnvZ[27] and validate an identical role in HK853. The presence of the product ADP increases this effect, supporting that the nucleotide promotes a competent conformation in the HK–RR complex for the phosphatase reaction. The structures reported here confirm that the nucleotide stabilizes the ATP-lid of HK853, which is involved in contacts with the RR loop (β3–α3) that follows and covers the phosphoacceptor Asp and which disposition is pivotal in the phosphotransfer/phosphatase reactions[40]. These peripheral interactions between loops in HK and RR which conformation is phosphorylation-state dependent have been proposed as selectors for conferring specificity in the reaction carried-out by the system[6,10]. The participation of the nucleotide mediated by the ATP-lid seems to be key for the phosphatase activity and becomes more evident at low pH where this activity is reduced. In this way, our experiments show that in the presence of nucleotide both HK853 and EnvZ present phosphatase activity even at pH 5. Liu and collaborators did not detect the phosphatase activity at pH 5[15] possibly because they used a version of HK853 lacking the ATP-binding domain while in the case of EnvZ the phosphatase activity was measured in the absence of nucleotide. These factors, together with the decrease in activity due to the protonation of the His, could lead Liu and collaborators to overestimate the effect of pH on the HK-mediated phosphatase activity, proposing the here refuted pH-gated mechanism for this activity.

In summary, our structural data, together with structures of HKs extracted from the PDB, show that the rotamer disposition of the catalytic His in HKs of the HisKA family is, in general, not regulated or influenced by the environmental pH, ruling out the pH-gated conformational switch to modulate the phosphatase activity as proposed by Liu and collaborators and puts in value the versatility of the histidine for molecular interactions and bioactivities. Finally, our results support that the correct arrangement of the HK and the RR, where contacting loops are influenced by the phosphorylation state of each protein and/or the presence of nucleotide, is key to generate an active center competent to carry out the phosphatase and, probably, the phosphotransfer reaction.

## Methods

**Protein production and purification.** Supplementary Table 3 lists the plasmid constructs used in this study, which were previously described. Site-directed mutagenesis in RR468 previously cloned in pET22b (Supplementary Table 3), was conducted to introduce mutation D53A using the Quickchange method (Stratagene) with the following primers FW: 5′-GATAGTTCTCGCCATAATGATGC CCGTG-3′ and RW: 5′-CATCATTATGGCGAGAACTATCAGGTCTGG-3′. The complete cytoplasmic portion of HK853 wild type and mutant H260A, RR468 wild type and mutant D53A, as well as EnvZ and the REC domain of OmpR were obtained following the procedures described[7,10,26,41]. Briefly, protein expression was performed in BL21-CodonPlus(DE3)-RIL (Stratagene) transformed with the appropriate plasmid (Supplementary Table 3). Cells were grown at 37 °C in LB medium supplemented with the corresponding antibiotics (33 μg ml$^{-1}$ chloramphenicol plus 33 μg ml$^{-1}$ kanamycin or 100 μg ml$^{-1}$ ampicillin) up to an OD$^{600}$ of 0.5–0.6, then, protein expression was induced with 1 mM isopropyl-b-ᴅ thiogalactopyranoside (IPTG) at 20 °C for 16 h. After induction, cells were harvested by centrifugation at 4 °C for 30 min at 3500 × $g$ and the pellet was resuspended in buffer A (50 mM Tris–HCl pH 8.0 and 150 mM NaCl) for Histag–HK853, Histag–HK853$^{H260A}$, Histag-RR468, and RR468$^{D53A}$ and buffer A′ (10 mM Tris–HCl pH 8.0 and 500 mM NaCl) for Histag-EnvZ and Histag-OmpR$_{REC}$ supplemented in both cases with 1 mM PMSF. Then the samples were sonicated and subsequently centrifuged at 16,000 × $g$ for 40 min. As HK853 and RR468 are thermophilic proteins, the collected supernatant after centrifugation for these proteins was incubated at 70 °C for 15 min and centrifuged again at 16,000 × $g$ for 40 min. Clear supernatants for the Histag proteins were loaded on HisTrap HP columns (GE Healthcare) pre-equilibrated with buffers A or A′. After washing with 10 column volumes of the corresponding buffer containing 50 mM imidazole, the Histag-protein was eluted by adding buffer supplemented with 350 mM imidazole. The Histag was removed from the eluted protein adding TEV protease at a molar ratio 1:25 (protease:eluted protein) and incubating for 16 h at 4 °C with slow stirring. After digestion, the sample was loaded one more time into the pre-equilibrated HisTrap HP column to separate the digested protein, non-retained in the column, from the Histag-protein and the protease. The non-retained protein was concentrated and loaded onto a Superdex prep grade HiLoad 200 16/60 or 75 16/60 (GEHealthcare) for the HKs or the RR, respectively, pre-equilibrated with buffer A. The eluted fractions were analyzed by SDS–PAGE and those fractions showing purest protein were selected, concentrated and stored at −80 °C.

In the case of RR468$^{D53A}$, which lacks of Histag, the supernatant after the second centrifugation was dialyzed against buffer B (50 mM Tris–HCl pH 8) at 4 °C with slow stirring. The dialyzed protein was loaded on a Mono Q 5/50 GL (GEhealthcare) column with buffer B for ion exchange chromatography. The protein was eluted using a gradient between 0 and 200 mM of NaCl in buffer B and all the fractions obtained were analyzed by SDS–PAGE and the purest fractions were concentrated and loaded onto a Superdex 75 prep grade HiLoad 16/60 pre-equilibrated with buffer B. After size exclusion, fractions with purest protein were selected, concentrated and stored at −80 °C.

**Autophosphorylation assays.** HK autophosphorylation at 0.12 mg ml$^{-1}$ was performed as described[10] in the presence of kinase buffer (100 mM Tris–HCl pH 8.0, 100 mM KCl, 10 mM MgCl₂) adding 0.1 μCi μL$^{-1}$ [γ-$^{32}$P] ATP and 0.1 mM ATP at different incubation times (1, 5, and 20 min). Autophosphorylation reaction at pHs different from 8 where carried out in kinase buffer substituting Tris–HCl by Bis–Tris at pH 5, 6, or 7. Samples were stopped by adding 2x loading buffer containing 50 mM EDTA and 4% SDS in 1:1 ratio and then were subjected to SDS–PAGE on 15% gels. Phosphorylated proteins were visualized by

autoradiography using a Fluoro Image Analyzer FLA-5000 (Fuji) and were evaluated with the MultiGauge software (Fuji).

To check stability for HK autophosphorylation, each HK at 2 mg ml$^{-1}$ was autophosphorylated with 0.1 μCi μL$^{-1}$ [γ-$^{32}$P] ATP and 0.1 mM ATP in kinase buffer containing 20 mM Tris–HCl pH 8 for 15 min. Then, phosphorylated HKs were loaded in a column with 0.5 ml of Sephadex G-25 fine (GE Healthcare) equilibrated with kinase buffer containing 100 mM of Tris–HCl pH 8 or 100 mM Bis–Tris pH 5 to remove the free ATP from the samples and interchange the pH buffer. After column, samples were incubated at room temperature, samples were stopped at 1, 3, 5, 15, and 30 min, subjected to SDS–PAGE and visualized as described previously.

**Phosphotransfer assays**. HK at 2 mg ml$^{-1}$ were phosphorylated with [γ-$^{32}$P] ATP in a buffer containing 20 mM Tris–HCl pH 8 during 15 min and free ATP was removed as described previously. Subsequently, phosphorylated HK were diluted in the corresponding buffer, 100 mM Tris–HCl pH 8 or 100 mM Bis–Tris pH 5 to 0.12 mg ml$^{-1}$ and incubated with 1.5 molar amounts (in terms of subunits) of RR468 and OmpR for HK853 and EnvZ, respectively, in the appropriate buffer for 1, 3, 5, 15, and 30 min. Shorter times <1 min were not collected for technical reasons. For EnvZ and OmpR, phosphotransfer assays were also performed at longer times (0.5, 1, 2, and 4 h) in order to observe phosphatase activity. When the effect of the nucleotide was analyzed 2 mM ADP was added to the reaction. EnvZ autophosphorylation at longer incubation times was also measured as a control. Samples were stopped as described previously, run in an SDS–PAGE gel, then, visualized and evaluated as described in the previous section.

**Phosphatase assay**. RR468 at 2 mg ml$^{-1}$ was phosphorylated for 1 h at room temperature with radioactive AcP[42], then, free AcP was removed using columns with 0.5 ml of Sephadex G-25 fine (GE Healthcare). Phosphorylated RR468 was diluted to 0.056 mg ml$^{-1}$ in the appropriate buffer (100 mM Tris–HCl pH 8 or 100 mM Bis–Tris at pH 5, 6, and 7) and mixed with equimolecular amounts of HK853 (0.12 mg ml$^{-1}$) diluted in the same corresponding buffer. When the effect of the nucleotide was analyzed 2 mM ADP was added to the reaction. Phosphorylated RR468 alone or mixed with HK853 in the absence or presence of ADP was incubated during 5, 15, 30, and 60 min. Samples were stopped as described previously, run in an SDS-gel, then, visualized and evaluated as described in the previous section.

**Crystallization, data collection, and model building**. Protein crystals were grown in sitting drops at 21 °C, by the vapor-diffusion approach. Crystallization trials were set up in the Cristallogenesis service of the IBV-CSIC from the conditions described by Podgornaia and collaborators for the HK853–RR68 complex[16] varying ammonium sulfate concentrations (from 1 to 2.2 M) along with 0.1 M sodium citrate at different pHs (5.5, 6.5, 7, and 7.5) or 0.1 M Bis–Tris at pH 5.3. The crystallization of the complexes between HK853–RR468$^{pHs}$, HK853$^{H260A}$–RR468$^{D53A}$, and HK853–RR468$^{D53A}$ was achieved using a mixture containing 10 mg ml$^{-1}$ HK853 and 7.5 mg ml$^{-1}$ RR468, (similar concentrations were used for the mutant variants), both in 50 mM Tris–HCl pH 8 and 150 mM NaCl adding 2 mM ADP. In the cases where RR468 wild-type was used 30 mM NaF, 5 mM BeSO$_4$, and 7 mM MgCl$_2$ was also added to the mixture in order to form BeF$_3$$^{-}$. Then, MRC2 plates were prepared dispensing 0.3 μL of the mixture over 0.3 μL of different mother liquour. For crystals HK853–RR468$^{pHs}$ obtained at pH 5.5, 6.5, 7, and 7.5 the mother liquor contained 1.8 M of (NH$_4$)$_2$SO$_4$ and 0.1 M sodium citrate at different pH 6.5, 7, and 7.5. For crystals of HK853–RR468$^{D53A}$, HK853$^{H260A}$–RR468$^{D53A}$ obtained at pH 7 and 5.3, respectively, the mother liquor conditions contained 1.8 M of (NH$_4$)$_2$SO$_4$ and 100 mM sodium citrate, while for the crystals grown at pH 5.3, the conditions were 1.2 M of (NH$_4$)$_2$SO$_4$ and 100 mM Bis–Tris pH 5.3. Cryoprotectant solutions for crystals grown at pH 5.5, 6.5, 7, and 7.5 were prepared exchanging the concentration of (NH$_4$)$_2$SO$_4$ to 2 M of lithium sulfate plus 0.1 M sodium citrate at the corresponding pH. For the crystals grown at pH 5.3, the cryoprotectant was prepared exchanging the concentration of (NH$_4$)$_2$SO$_4$ to 1.8 M lithium sulfate plus 0.1 M Bis–tris pH 5.3. Before crystal diffraction, crystals were soaked rapidly in the cryoprotectant solution prior to flash cooling in liquid nitrogen. Crystals were diffracted at beamlines I04 in Diamond light source synchrotron (Didcot, UK) and beamline BL13-XALOC in Alba Synchrotron (Cerdanyola del Vallès, Spain) and the datasets showing the highest resolution were used to solve the structures. The data integration and reduction was processed with Mosflm[43] and Aimless[44,45] programs from the CCP4 suite[45]. For molecular replacement, the previously reported structure of the HK853–RR468 complex crystallized at pH 5.6 (PDB: 3DGE[10]) was used as the search model with the program Phaser[46]. The structural model was iteratively built by COOT[47] and refined by Refmac[48]. Data collection and model refinement statistics are given in Table 2 for complexes containing with wild-type proteins and in Table 3 for complexes containing mutant variants.

Figures were produced using USCF Chimera[49] and superpositions were performed using programs from CCP4 suite[45].

**Reporting summary**. Further information on research design is available in the Nature Research Reporting Summary linked to this article.

## Data availability
The X-ray crystallographic coordinates reported for all the structures have been deposited at the Protein Data Bank. For HK–RR$^{pHs}$, PDB accession codes are 6RGY for pH 7.5, 6RFV for pH 7, 6RGZ for pH 6.5 and 6RH0 for pH 5.5. For the mutant HK853–RR468$^{D53A}$, PDB accession codes are 6RH1 for pH 7.5 and 6RH2 for pH 5.3 while for the mutant HK853$^{H260A}$–RR468$^{D53A}$, PDB accession codes are 6RH7 for pH 7.5 and 6RH8 for 5.3. Source Data underlying Figs. 4–6, as well Supplementary Figs. 4 and 5 are provided as a Source Data file. The authors declare that all other relevant data supporting the findings of this study are included in this published article and its Supplementary Information files, or from the corresponding authors upon request.

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

## Acknowledgements

We would like to thank the IBV-CSIC Crystallogenesis Facility for protein crystallization screenings. The X-ray diffraction data reported in this work were collected in experiments performed at BL13-XALOC and I04 beamlines at ALBA (Cerdanyola del Vallès, Spain) and DLS (Didcot, UK) Synchrotrons, respectively. We thank Local Contact and staffs of the beamlines for providing assistance during data collection. The following funding is acknowledged: Spanish Government (Ministerio de Economia y Competitividad; Grant no. BIO2016-78571-P to Alberto Marina; grant No. BFU2016-78606-P to Patricia Casino; contract No. RYC-2014-16490 to Patricia Casino); Valencian Government Prometeo program (grant No. II/2014/029 to A.M.). C.M.-M. is the recipient of a Ph.D fellowship from the Programa de Becas, Secretaría de Educación Superior, Ciencia, Tecnología e Innovación of Ecuador Government (2015-AR2Q9228). X-ray diffraction data collection was supported by Diamond Light Source block allocation group (BAG) Proposal MX14739 and MX20229 and Spanish Synchrotron Radiation Facility ALBA Proposal 2017072262 and 2018072901.

## Author contributions

Conceptualization, A.M. and P.C.; Methodology, A.M., P.C. and C.M.-M.; Investigation, C.M.-M., L.M.-R., A.F.-R., P.C. and A.M. Writing original draft, A.M. and P.C.; Funding acquisition, A.M. and P.C.; Resources, A.M., P.C., C.M.-M. and L.M.-R.; Supervision, A.M. and P.C.

## Competing interests

The authors declare no competing interests.
