## [Peer Review File · Nature Communications]

Reviewers' comments:

Reviewer #1 (Remarks to the Author):

For the authors

I found the article of great and broad interest. The article addresses a fundamental question of the Histidine Kinase Superfamily and Two-Component System field: the reaction mechanisms of the three related enzymatic activities, responsible for the cell signaling of environmental cues sensing in Prokaryotes. The treatment to the topic is very original.

The article's focus, as it is written at the moment consist in ruling out the pH-gated model of phosphatase activity of HisKA histidine kinases, a model published two years ago. I think this conversation between labs visions is interesting and amusing for the readers (it keeps us awake), but should not be the focus. The article has a great value by itself.

In the article not only own data is presented, but also it includes a big piece of analysis of deposited pdb coordinates. This is a strong aspect of the article, because it implies to put the results in context of previous data.

The amount of data presented in the paper is large enough for publication, however one more effort at the writing stage is needed to achieve a nice, focused, concise and clear article. Apart from several controls and adjustments on some experiments, stated bellow, all the experimental work is practically done.

The introduction is very clear. However, I have a few comments.

It would be appropriate to introduce the autophosphorylation and the phosphotransferase activities and their mechanisms, apart from the phosphatase, as they are also studied in the article (Results Section). Figure 1 can include not only the phosphatase reaction mechanism, but also the autophosphorylation and phosphotransfer reactions.

According to the CpxA – Mechaly, A et al, 2014- and chimeric EnvZ -Casino P at all, 2014- Michaelis complex papers, the autophosphorylation of the His residue occurs exclusively in N ϵ . Do the authors agree? It seem to be an important piece of information. In figure 1A either N ϵ and N γ are shown phosphorylated. Why is that?

What I learned from the article (together with the CpxA and HK853(EnvZ)CHIM, 2014- Michaelis complex papers) if I understood well, is that the trans rotamer of the phosphorylatable His is required not only for the phosphatase activity (as proposed by Liu Y. at all, 2017) but also for the other two enzymatic activities, while the gauche- is an inactive conformation. If that is the case it should be clearly stated in this Section, in order to follow the story more easily and to avoid some confusing point at the Results Section (see bellow).

The article findings are the following:

- The rotamer disposition of the phosphorylatable His is not influenced by the environmental pH in different HK structures belonging to the HKA family already deposited in the PDB, ruling out a pH-gated model.
- The rotamer disposition of the phosphorylatable His is not influenced by the environmental pH in HK853 at different pHs, published here, reinforcing the previous observation.
- Phospho-His can be stabilize in a gauche rotamer, as an inactive resting state.
- The phosphatase activity decreases at low pH, as well as autophosphorylation both for HK853-RR468 and EnvZ-OmpR systems.
- The phosphotransferase activity for the authors is independent of the pH. (This should be revised, see bellow).

The Results, in general are well organized. The subheadings are informative of the content. However, each subsection should finish with a clear conclusion. Besides, I have several concerns and comments.

Crystal structure analysis. The authors assumed that the sulfate ion in the vicinity of the His residue mimics a phosphate group. Although the sulfate is actually closer than the phosphate group in the Michaelis complex of CpxA and the chimeric EnvZ, the distance is larger than the 1.7Å or so, expected for the covalent bond N-P. The mentioned assumption, should be grounded. Phosphatase activity. The experiments should include a control of phosphorylated RR in the absence of HK for both systems.

Phospho-HK stability at pHs 8 and 5. In Figure 5. I see that the intensity of the band corresponding to the HK853-P at time 0 at pH 8 and 5 differ a lot in both b and c. Why is that? Is it because the dephosphorylation of HK853 at pH 5 is so fast or is it an issue of protein stability? Do you have a charge control of those gels (Coomassie staining or similar)? This could be included for all the gels, but is especially important here. Maybe be circular dichroism spectra and T_m estimation could be useful as stability controls.

pH effect on phosphotransferase activity. I am not convinced of the independence of pH of phosphotransferase activity because the time range assayed is not adequate to see differences. The experiments (which correspond to Figure 5c and Figure 6c) should be adjusted in order to evaluate the evolution of phosphotransferase activity. Actually, I understood that the authors expected a pH dependence similar to the one observed for the autophosphorylation and the phosphatase activities. Is that right?

In the Discussion Section the authors mentioned that

The structures reported here confirm that the nucleotide stabilizes the ATP-lid of HK853, which is involved in contacts with the RR loop ($\alpha 3$ - $\beta 3$) that follows and covers the phosphoacceptor Asp and which disposition is pivotal in the phosphotransfer/phosphatase reactions.

This analysis should be developed in the Results Section, maybe under a separate subheading. Do other deposited pDBs show the same?

Other comments.

The Article Title. Revisiting the pH-gated conformational switch on the activities of HisKA-family histidine kinases

I think pH should not be in the title, because I see after going through the article that the pH is more an experimental tool, than a parameter sensed by the histidine kinase. The title could be something like this: "Revisiting the reaction mechanisms of the enzymatic activities of HisKA-family histidine kinases".

The Abstract. It is not stated which kind of experiments have been made. The only message is that the pH-gated model for the phosphatase activity has been ruled out.

In page 10 the authors said

In order to answer this question we analyzed if the rotamer could be induced by side-chain interactions or reflected a state of minimum energy.

Then which of these two possibilities actually occurs is not clear stated.

Page 12. This argument is not clear

Moreover, at pH 5 the His is a better phosphate donor than at basic pH given the weakening of the P-N bond, counteracting a possible slight loss of nucleophilicity that the Asp could experience when approaching its pKa.

In materials and methods. It is not clear if a crystallization screening was performed or the authors went directly to a known crystallization condition.

In Figure 6. b. The background corresponding to time 15 at pH 8 is much higher than the others. Why is that? If the different aliquots along time were taken from a single reaction mixture then the background should not differ, should it? c. The lane corresponding to time 0 comes from a different gel. I encourage the repetition of this experiment. Additionally, the same comment as previous figure, why the band at time 0 at pH 5 is less intense than at pH 8.

Crystallographic tables 2 and 3. Rmeans and CC1/2 (%) values should be added.

Other minor comments.

In general. The proper way to name the rotamers I believe is trans/gauche- $\chi 1$ rotamer.

In some the figures. The net charge of sulfate (SO_4^{2-}) is missing in the figures.

In the Results Section. Homogenize subheading with and without stops.

Page 6. The inactive/active terminology is not totally clear. In this case I believe it corresponds to the phosphatase activity. It should be stated clearly. In the introduction there is a similar situation but there it is unmistakably indicated. (Overall, in the pH-gated model the authors proposed that the transition between inactive to active states for the phosphatase activity ...).

The pH-gated model proposes that in HKs of the HisKA family the rotamer disposition of the phosphorylatable His is regulated by pH, acquiring, an inactive gauche- rotamer at pH between 5.2-6.5 and an active trans rotamer at pHs above 6.5 (Fig 1b).

Page 6. The fact that this pdb corresponds to the isolated HAMP DHP should be informed in the text.

Finally, contrary to the proposal of the pH gated model the structure crystallized at the lower pH (4.6) corresponding to the isolated EnvZ HAMP DHP.

Page 6. I have three comments on this sentence.

Indeed, in those cases where the phosphorylatable His is trapped performing the autophosphorylation reaction, as in subunit A of CpxA (PDB: 4BIW8, solved at pH 8.5) and subunit A of chimeric version of chimeric EnvZ (PDB: 4KP47, solved at pH 7.5), or the phosphatase competent reaction in HK853 (PDB: 3DGE10, solved at pH 5.6), it shows an invariably trans rotamer, that is for an active state.

The fact that this pdb corresponds to a chimeric EnvZ should be informed in the text.

This sentence seem to be stating the opposite to the previous one. The isomerization of the phosphorylatable His is related with the catalytic state or not?

"..., that is for an active state." It is not clear.

Page 10. The citation of the previously described stability should be included.

Notice that as it was previously described, the phosphorylated RR was stable at the tested pH range (8-5).

Page 14. This sentence does not belong to the section. Maybe could be moved to Discussion section.

Overall, our in vitro functional assays confirm that, as in the case of HK853, the autophosphorylation and phosphatase reactions are impaired at pH 5 while the phosphotransfer activity is unaffected at this pH, in close relation with the chemistry of the catalytic residues involved in each reaction but not with a side-chain conformational change of the phosphorylatable His as it is proposed in the pH-gated model.

Figure 1. The atom name labels to the His should be included.

Figure 2. a and c have their zoom in (d and e). b could have zoom in, too.

Figure 4. b. The frame of the pH 5 experiment is thicker. In this case ADP- and ADP+ experiments are in different gels. Maybe be it is better to indicate it clearly with a line. So nobody could say it was confusing. For the other pHs (8, 7 and 6), ADP- and ADP+ experiments are in the same gel. Is that correct? In the graph there are no error bars in the graph. Why? Is it because the sign of the plot are very big.

In the legend of Fig. 1. Two distinct rotamers trans and gauche- are shown for the catalytic His, where just the trans rotamer is involved in the phosphatase reaction.

In the legend of Fig. 4. The title is confusing. It would be better in my opinion Phosphatase activity of HK853 on RR468. In b it should be mentioned that the gel corresponds to a representative experiment of the three repetitions. It should be mentioned that the mean values of three independent experiments are plotted in the graph. The bars correspond to the standard deviation.

Crystallographic tables 2 and 3.

Total nro. of reflexions.

Add unit to the B factor.

Average B factor (\AA^2). PDB code

Supplemental table S2. The rotameric form of the structures reported here should be checked.

They should be all gauche-, shouldn't they?

Reviewer #2 (Remarks to the Author):

The manuscript by Mideros-Mora et al studies effects of pH on the structure and function of the histidine kinase HK853. By solving and comparing a series of HK853-RR468 complex structures at different pH, the authors argue against a pH-gated model in which the pH determines the trans- or gauche-position of the phosphorylated His residue and dictates the active or inactive state of

HK853. The authors further claim that the pH-dependent activities of HK853 are due to different protonation states of the His residue based on biochemical analyses of various HK activities at different pH. It is not a novel concept that protonation of the His at different pH impacts the enzyme activities. Moreover, biochemical results presented in this manuscript appear not sufficient to favor or disfavor the His protonation model. A pH-dependent conformation-switching model cannot be completely excluded.

The following are detailed comments:

1. All of the WT complex structures have two phosphoryl group mimics, a BeF₃ and a sulfate ion, suggesting that both HK and RR proteins are phosphorylated. But none of the kinase, phosphotransfer or phosphatase reactions will result in a complex with both proteins phosphorylated. Are such complexes physiologically relevant? Will the pH-independence of this non-reacting resting state be readily translatable to other relevant structures?
2. The main biochemical evidence against the pH-gated switch model is that pH did not affect the phosphotransfer reaction. However, data shown in Fig. 5C did not support the author's claim that phosphotransfer is independent of pH. At pH 5, initial level of HK853~P was less than that at pH 8 but it took a longer time (~15 min) than at pH 8 (1 min) to reach a similar level. This suggests that the reaction rate is lower at pH 5 than at pH 8. Quantification of phosphorylation is required to draw any conclusion about reaction rates.
3. As described above, the kinase, phosphotransfer and phosphatase activities shown in Fig. 4 and 5 are all reduced at acidic pH. These results cannot differentiate a His protonation model or a conformational switch model. pH-dependent conformational dynamics could occur at other residues and still contribute to the activity difference. The F19 NMR studies in Liu et al suggested the switching of conformation states at a time scale different from the His protonation.
4. A SO₄ ion can occupy the same position as a PO₃ group. Will the different geometry or position of oxygen atoms in PO₃ allow similar residue interactions as SO₄ to stabilize the gauche position? Is it possible to model a phosphorylated His residue onto the current structure to explore the potential residue interactions?
5. Fig. 1a, it is generally believed that the His phosphorylation in HKs occurs at N3 position. Highlighting the N3 phosphorylated form may reduce any potential confusion.
6. Fig. 4a, at pH 5, the phosphorylation band intensity at 60 min appears much less than corresponding lanes at higher pH values. Does this suggest that the RR autophosphorylation rate is also affected by pH?
7. Page 11, paragraph 1, the last sentence. "lowering the pH below 6 increases the His protonation..." The wording here imply pH 6 as a threshold value, but neither the protonation mechanism nor the observed data supports the presence of such threshold.
8. Page 13, paragraph 2, line 9. Typo, "high-life" change to "half-life".

Reviewer #3 (Remarks to the Author):

The manuscript by Mideros-Mora et al revisits the model recently published by another group (Liu et al. 2017, Nat commun), who proposed that the phosphatase activity of the histidine kinase HK853 is negatively regulated at acidic pH through a switch in the rotamer disposition of the side chain of the catalytic histidine residue, which precludes the coordination of the water molecule. In the present report, the authors show that:

- i) in 24 other reported structures of histidine kinases of the HisKA family, there was no correlation between the pH used for crystallization and the rotamer disposition (gauche or trans)
- ii) the structure of the HK853-RR468 complex in different pH conditions invariably displays a gauche rotamer disposition of the histidine, ruling out a pH-induced change in rotamer disposition; the structures support the hypothesis that the gauche disposition represents, independently of the pH, the resting state for the His awaiting either the RR for phosphotransfer, or the phosphorylated RR for dephosphorylation.
- iii) they confirmed in vitro the previous observation that the phosphatase activity of HK853 over phosphorylated RR468 drops with decreasing pH. However, they observed that this pH sensitivity

was much lower in the presence of ADP. Together with the fact that the previous observation by Liu et al was done using a HK853 variant lacking its catalytic and ATP binding (CA) domain, this suggests that the CA domain stabilizes the HK in a conformation competent to interact with the RR.

iv) they showed that the autokinase activity is also inhibited at low pH, but not the phospho-transfer activity of HK853, although it also involves the same His residue.

Similar results were obtained using the EnvZ-OmpR two-component system in vitro.

As both the kinase and phosphatase activities depend on the ionization state of the histidine residue, they propose that the pH sensitivity of the kinase and phosphatase activities is due to an increased protonation of the histidine at low pH, thus decreasing its capacity to act as a general base for activation of the catalytic water in the phosphatase reaction, or as a nucleophile to attack ATP in the kinase reaction. In contrast, the phosphotransfer reaction depends on the Asp residue of the RR, which protonation is weakly affected at pH 5.

Altogether, the conclusions are well supported by the presented data and provide convincing evidence that the previously proposed pH-gated model is invalid. This finding is of interest for the community in the field of TCS.

My major concern is that the authors do not sufficiently comment on the discrepancies between their observations and those made by Liu et al regarding the conformational changes they observed at low pH. I have a few minor comments/corrections:

P2 L8: phosphorylatable

P3, 2 lines from the bottom: ...similar to the one observed...

P5, L2: ...was based on...

P5, 2 lines from the bottom: the effects

P6, L10: 18 or 20 ?

P7, L4: The proposed pH-gated model (ref 15) was structurally...

P9, 12 lines from the bottom: ...phosphorylatable Asp due to its mutation...

P10, L1: ...to the one observed...

P10, L5: ...an equivalent position as HK853.

P10, title: Effect of pH on the phosphatase...

P11, 13 lines from the bottom: ...the in vitro assays support that the...

P13, L11: The pH-gated model was also proposed on the basis of the pH-dependent phosphatase...

P13, 7 lines from the bottom: ...a similar result as in HK853...

P17, L4: EnvZ not in italics

P17, L9: ...were carried out in...

P17, last line: appropriate

P21, L10: HK853

P21, L13: phosphorylatable

P21, 5 lines from the bottom: ...the same code as in a).

P21, 3 lines from the bottom: as sulfate

L22: Fig. 4. RR468 autophosphorylation

L22, last line: delete normalizing

L23, L5: autophosphorylation with

Table 1: what does the * means (His rotamer for each subunit in dimer*)

Legend to Fig. S1, L8: as well as

Reviewer #4 (Remarks to the Author):

In this manuscript, Mideros-Mora et al. present structural and biochemical data challenging a recently proposed mechanism by which pH regulates the phosphatase activity of histidine kinases (HKs) from the HisKA family (Liu et al., 2017). This mechanism postulates that the catalytic histidine adopts a gauche- rotamer at low pH, abolishing phosphatase activity. By analyzing X-ray

structures of prototypical HisKA HKs (HK853, EnvZ, CpxA, VicK and Walk), solved at various pHs, Mideros-Mora et al. show that the rotameric state of the catalytic residue and the pH of the crystallization conditions are not clearly correlated. The authors also report structures of the HK853-RR468 complex determined at different pHs. In all of them the same rotamer (gauche-) is observed. In addition, they performed in vitro assays that corroborate the inhibitory effect of low pH on the phosphatase activity of HK853 and EnvZ. Importantly, the authors also characterized the autokinase and phosphotranfer activities, detecting that only the former is affected by changes in the pH value. From these results, they conclude that is the chemistry of the histidine and not the rotamer that adopt, what dictates the activity levels.

In my opinion, this convincing study represents an important contribution to field of bacterial signaling. However, I do have some comments, which should be addressed before publication of the manuscript.

(i) The PDB entry 4CTI (corresponding to a chimera between the HAMP domain of Af1503 and the DHp and CA domains of EnvZ) is missing in Table 1. Some other structures might also be included in the table. For instance, the PDB entries 5UKV, 4MT8 or 6DK7. I guess that the structure of the ThkA-TrrA complex was not included in the analysis because of its low resolution.

(ii) Various of the papers reporting the crystallization conditions shown in Table 1 are not cited in the manuscript.

(iii) CC1/2 values should be reported in Tables 2 and 3.

(iv) The authors suggest that some of the structures represent a resting state prior either phosphotransfer or dephosphorylation. I would also consider another possibility, a snapshot right after the dephosphorylation reaction, with the sulfate ion mimicking the leaving Pi molecule.

(v) Figure 4. I assume that the phosphatase assays were carried out using radioactive AcP. In that case, it should be mentioned in the Results and Methods sections.

(vi) Did the authors perform phosphotransfer assays like those shown in Figure 5c, using the phosphatase-defective mutant T264A instead of wild-type HK853? A priori using this mutant the phosphotranfer assays wouldn't be affected by the strong phosphatase activity of HK853.

Reviewers' comments:

Reviewer #1:

It would be appropriate to introduce the autophosphorylation and the phosphotransferase activities and their mechanisms, apart from the phosphatase, as they are also studied in the article (Results Section).

Following the indications of the reviewer we have included additional information in the introduction describing the current knowledge about the mechanisms proposed for the autophosphorylation and phosphotransfer activities. The new paragraph in the introduction (page 3) reads: *“the kinase reaction has been visualized in the crystal structures of two HisKA HKs, CpxA and a chimeric version of EnvZ, both trapped catalyzing this reaction where the His acts as a nucleophile to attack the γ -phosphate of ATP helped by its interaction with a neighbored conserved acidic residue that acts as a general base to increase the His nucleophilic character^{7,8}. Secondly, the phosphoryl group is transferred from the His to the conserved Asp residue at the REC domain in the phosphotransfer reaction⁶. This reaction has not been visualized yet for the HisKA family but the structure of the complex between a mutant version (H188E) of the HK DesK, a member of the minority HisKA_3 family, with the REC domain of DesR⁹ has been proposed as the conformational state of this reaction. However, the absence of a phosphoryl group or its mimetic in the structure hinders a detailed deduction on the phosphotransfer mechanism.”*

Figure 1 can include not only the phosphatase reaction mechanism, but also the autophosphorylation and phosphotransfer reactions.

We thank the reviewer for the suggestion that improves the manuscript. We have added in Figure 1b the two additional reactions, phosphatase and phosphotransfer, catalysed by the system.

According to the CpxA – Mechaly, A et al, 2014- and chimeric EnvZ -Casino P at all, 2014- Michaelis complex papers, the autophosphorylation of the His residue occurs exclusively in N ϵ . Do the authors agree? It seem to be an important piece of information. In figure 1A either N ϵ and N γ are shown phosphorylated. Why is that?

We thank the reviewer for the comment. Indeed, the structures of CpxA and chimeric EnvZ trapped in the conformation of autophosphorylation showed that N ϵ was the attacking atom to become phosphorylated. However, in Figure 1a we wanted to highlight the versatility of the His to become phosphorylated in one or the other N atom as it has been observed for other enzymes kinases. To be more clear we have indicated in the figure that N ϵ phosphorylation is occurring in HKs and we have introduced this information in the figure legend adding the following sentence *“Structural and functional studies on HKs support autophosphorylation at N ϵ ^{7,8,49} while autophosphorylation at N γ happens at other enzymes such as nucleoside diphosphate kinases⁵⁰”*

What I learned from the article (together with the CpxA and HK853(EnvZ)CHIM, 2014-Michaelis complex papers) if I understood well, is that the trans rotamer of the phosphorylatable His is required not only for the phosphatase activity (as proposed by Liu Y. at all, 2017) but also for the other two enzymatic activities, while the gauche- is an inactive conformation. If that is the case it should be clearly stated in this Section, in order to follow the story more easily and to avoid some confusing point at the Results Section (see bellow).

The article findings are the following:

- The rotamer disposition of the phosphorylatable His is not influenced by the environmental pH in different HK structures belonging to the HKA family already deposited in the PDB, ruling out a pH-gated model.
- The rotamer disposition of the phosphorylatable His is not influenced by the environmental pH in HK853 at different pHs, published here, reinforcing the previous observation.
- Phospho-His can be stabilize in a gauche rotamer, as an inactive resting state.
- The phosphatase activity decreases at low pH, as well as autophosphorylation both for HK853-RR468 and EnvZ-OmpR systems.
- The phosphotransferase activity for the authors is independent of the pH. (This should be revised, see bellow).

We thank the reviewer for synthesizing the findings of the manuscript. We consider them really accurate, thus we have taken the liberty to include them at the end of the introduction section as a summary of the manuscript.

Regarding the influence of the pH in the reactions, our results agree with the chemistry of the residues involved in each specific reaction. His has a main participation in kinase and phosphatase reaction and its sidechain pKa is around 6.0. Therefore, changing from pH 8.0 to 5.0 will affect this residue that will pass from completely unprotonated to almost completely protonated, loosing its capacity to work as a general base or as a nucleophile required to catalyze the reactions. However, in the phosphotransfer reaction the Asp residue also contributes playing a major role. Since the Asp sidechain pKa is around 3.9-3.65, at pH 5 will be almost completely unprotonated (~ 90%), as at pH 8.0, thus this reaction will be less influenced by the pH change.

The Results, in general are well organized. The subheadings are informative of the content. However, each subsection should finish with a clear conclusion.

As the reviewer has suggested, the end of each section contains now a conclusion sentence that summarizes the findings.

Crystal structure analysis. The authors assumed that the sulfate ion in the vicinity of the His residue mimics a phosphate group. Although the sulfate is actually closer than the phosphate group in the Michaelis complex of CpxA and the chimeric EnvZ, the distance is larger than the 1.7Å or so, expected for the covalent bond N-P. The mentioned assumption, should be grounded.

We agree with the reviewer that the sulfate ion cannot completely emulate the phosphoryl group since it is not covalently bound to the His. However, the sulfate is a divalent anion analogous to phosphate that tends to occupy phosphoryl-binding sites providing mechanistic information about the ground-state of phosphorylated-mediated reactions (Karen A (Editor), 2018, Phosphatases, Methods in enzymology, 607: 157-181). Therefore, it is widely assumed that sulfates usually occupy positions corresponding to phosphates or phosphoryl groups in phosphorylated macromolecules, having multiple examples deposited in the PDB and we indicate this fact in the manuscript (page 10): *“as it is known, sulfate ions tend to occupy the position of phosphates in crystal structures mimicking its actions in several biological processes such as protein phosphorylation. In this way, sulfate ions have been found occupying the position of the phosphoryl group in phospho-Ser/Thr^{22,23}, phospho-His²⁴ or phospho-Asp^{10,25} among others”*. In the case of HKs, a sulfate ion interacting with the phosphorylatable His has been observed not only in the structures presented in the manuscript but also in other structures of HKs solved alone (HK853 PDB:2C2A; Walk PDB:5C93) or in complex with its cognate RRs (HK853-RR468 PDB: 4JAV), were the His was showing also a *gauche*- rotamer. These evidences ground our assumption that the sulfate ion interacting with the His is located in a phospho-His binding site and not

in a nonspecific site. Indeed, the structures presented in the manuscript show that when His was mutated to Ala the sulfate was absent, indicating a specific binding.

Phosphatase activity. The experiments should include a control of phosphorylated RR in the absence of HK for both systems.

As the reviewer requires, we have added at Fig 4b the control of RR468 phosphorylated along the course of the reaction.

Unfortunately, we cannot include a similar control for OmpR since we (and other) were unable to phosphorylate OmpR by acetyl phosphate (AcP). Thus, we performed experiments using phosphorylated EnvZ to observe phosphotransfer and phosphatase activity over OmpR (Fig 6c). Phosphotransfer experiments produced phosphorylated OmpR that was observed even at longer incubation times, up to 4h (Supplementary Fig. 6), since the phosphatase activity of EnvZ is lower than that of HK853. The presence of ADP, stimulates the phosphatase activity of OmpR at both pHs but in a higher extent at pH 8 (Fig. 6c and Supplementary Fig. 6c).

Phospho-HK stability at pHs 8 and 5. In Figure 5. I see that the intensity of the band corresponding to the HK853-P at time 0 at pH 8 and 5 differ a lot in both b and c. Why is that? Is it because the dephosphorylation of HK853 at pH 5 is so fast or is it an issue of protein stability? Do you have a charge control of those gels (Coomassie staining or similar)? This could be included for all the gels, but is especially important here. Maybe be circular dichroism spectra and Tm estimation could be useful as stability controls.

Following the indications of the reviewer we have checked the stability of the proteins at pHs 8 and 5 and we have performed charge control gels. The circular dichroism spectra for HK853 and EnvZ (Supplementary Fig 4a) at acidic and basic pHs, as well as for RR468 and OmpR (shown below), shows similar profiles independently of the pH. The Coomassie stained of charge control SDS-gels confirm that the amount of protein for HK853 and EnvZ was not changing along the experiment of autophosphorylation stability (Supplementary Fig. 4b). Therefore, the differences observed at time 0 between pHs are not due to protein stability.

The differences in phosphorylated HK (P-HK) observed in the gels may be due to the low stability of the P-N bond at low pH. As the experiment confirms, the half-life of P-HK853 and P-EnvZ are, respectively, around 10 and 5 times lower at pH 5 than pH 8. To carry out the P-HK stability assays we must to remove the residual ATP after HK autophosphorylation by using a gel filtration. This purification step takes around 10-20 min (notice that half-lives for P-HK853 and P-EnvZ at pH 5 are around 10 min), therefore the starting amount of P-HK is reduced around 50-75 % at pH 5 vs pH 8 since we load identical amounts of protein in the gels, explaining the lower intensities for the bands observed at low pH. We have normalized the phosphorylated signal at

each time point versus time zero (elution of the column) at the corresponding pH for the calculation of half-lives.

pH effect on phosphotransferase activity. I am not convinced of the independence of pH of phosphotransferase activity because the time range assayed is not adequate to see differences. The experiments (which correspond to Figure 5c and Figure 6c) should be adjusted in order to evaluate the evolution of phosphotransferase activity. Actually, I understood that the authors expected a pH dependence similar to the one observed for the autophosphorylation and the phosphatase activities. Is that right?

At the pH range analysed (8 to 5), we expected that kinase and phosphatase activities were more dependent of pH than phosphotransferase. This assumption was based on the effect of the pH over the chemistry of the residues involved at each reaction, the His for kinase and phosphatase and the His and Asp for the phosphotransfer.

Following the reviewer's indications, we performed and quantified new phosphotransfer experiments with HK853 and EnvZ at similar incubation times (30 min). The quantification shows that at both pHs more than 95% and 85 % of the phospholabelled band has been transferred at the first time point (1 min) for HK853 and EnvZ, respectively. In all cases, a reminiscent small amount of P-HK is observed whose dephosphorylation kinetic is much slower. This band could be due to the fact that we perform the tests in a ratio 1:1 (HK:RR). Accurate protein quantification of the proteins is not perfect, especially for the RR which has a low molecular weight, so there may be some excess of HK over RR. At pH 8, this is not a problem since HK853 preserves the phosphatase activity and the P-RR is dephosphorylated so new cycles of phosphotransfer/phosphatase are produced, eliminating all this P-HK excess. In the case of pH 5.0, phosphatase activity is very low, even more when we have eliminated ADP, thus this reminiscent phospholabelled HK takes longer to disappear. In fact, the instability of the P-His at pH 5 ($t_{1/2}$ 10 min) is also contributing to this low rate of eliminating the remaining P-HK853. In the new figure 5c we have repeated the experiments using a HK853:RR468 ratio of 1:1.5 and we have confirmed that almost 98 % of the phosphoryl group is transferred at 1 min, supporting that the phosphotransfer reaction is not affected or affected to a lesser extent than phosphatase and kinase by pH, which are highly impaired at pH 5. In any case, we agree with the reviewer that any reaction involving catalytic residues which protonation state can change within physiological pH is dependent on the pH. However, we believe that our assumption was correct since the phosphotransfer activity is much less affected by pH (at the range 8 to 5) than kinase and phosphatase reactions, supporting that the phosphorylatable Asp plays an important role in this reaction being less likely to be influenced by pH 5 due to its low pKa (3.9-3.65). Meanwhile, the autophosphorylation and phosphatase activities are more affected by pH due to its influence on the ionization state of the catalytic His. We have clarified this point as follows (Page 14): *"In this way, the phosphotransfer reaction seems to be more independent of pH at the range assayed (from 8 to 5), probably because the hydrolysis of the P-N bond and the phosphoryl transfer is promoted by the attack of the phosphorylatable Asp acidic group which protonation state is weakly affected at pH 5.0 since its pKa is ~3.9. Moreover, at pH 5 the phospho-His may be a better leaving group than at basic pH given the weakening of the P-N bond due to its propensity to become protonated whereas Asp will still be in its ionized form to catalyse phosphoryl transfer."*

In the Discussion Section the authors mentioned that the structures reported here confirm that the nucleotide stabilizes the ATP-lid of HK853, which is involved in contacts with the RR loop (α 3- β 3) that follows and covers the phosphoacceptor Asp and which disposition is pivotal in the phosphotransfer/phosphatase reactions. This analysis should be developed in the Results Section, maybe under a separate subheading. Do other deposited pDBs show the same?

The structures presented in the manuscript show a HK853 with an ADP molecule bound in its nucleotide binding site and RR468 showing a phosphorylated conformation induced by the phosphomimetic BeF_3^- (BeF). The phosphorylated conformation of RR468 is confirmed by the structural comparison with free RR468 in its apo form (PDB 3DGF) or in complex with BeF (PDB 3GL9), or by the structural comparison with many other RRs which structure has been solved in the phosphorylated and unphosphorylated conformations. It is characteristic that loops $\beta 3-\alpha 3$ ($\text{L}\beta\alpha 3$) and $\beta 4-\alpha 4$ ($\text{L}\beta\alpha 4$) of REC domains acquire an “open” conformation in the phosphorylated state, meanwhile in the unphosphorylated state these loops are in “close” conformation approaching the phosphorytable Asp. Indeed, the open conformation of $\text{L}\beta\alpha 3$ is induced and/or stabilized by extensive polar contacts between the phosphoryl group (mimic in our structure by BeF) and the main chain of residues I54 and M55, which also make contacts with the Mg chelated by the phospho-aspartic. In this conformation, $\text{L}\beta\alpha 3$ contacts the HK853 ATP-Lid through the sidechains of M55 and P57 with E438 and Y437 (see figure). The “open” conformation of RR468 $\text{L}\beta\alpha 3$ and $\text{L}\beta\alpha 4$ is required to interact with HK853, since superimposition of unphosphorylated RR468 (apo form PDB 3DGF; magenta in the figure) on RR468 of the complex (pale green in the figure) shows clashes with the ATP-lid and DHp domain (see figure).

These observations, exposed in the discussion of the manuscript, reinforce our previous proposal that HK and RR recognize the phosphorylation state of their partner (Casino et al, Cell, 2009). In this way, the phosphorylated HK recognizes the dephosphorylated RR for the phosphotransfer reaction, while the dephosphorylated HK recognizes the phosphorylated RR in the phosphatase reaction. This proposal needs to be confirmed experimentally and it represents a current line of work in our laboratory. Therefore, we consider that making a more exhaustive analysis of these data and move them to the results section is: i) too premature, ii) would require additional experimentation that would justify and independent publication, and iii) is outside of the general scope of this manuscript. Therefore, we would prefer to leave this message in the discussion section. We have reflected this fact in the manuscript including the following sentence in the discussion section (Page 17): “*These peripheral interactions between loops in HK and RR which conformation is phosphorylation-state dependent have been proposed as selectors for conferring specificity in the reaction carried-out by the system*^{6,10}”

Other comments.

The Article Title. Revisiting the pH-gated conformational switch on the activities of HisKA-family histidine kinases.

I think pH should not be in the title, because I see after going through the article that the pH is more an experimental tool, than a parameter sensed by the histidine kinase.

The title could be something like this: "Revisiting the reaction mechanisms of the enzymatic activities of HisKA-family histidine kinases".

The manuscript tends to revise the regulatory model raised by Liu Y et al. in Nature Communications 2017 entitle "A pH-gated conformational switch regulates the phosphatase activity of bifunctional HisKA-family histidine kinases". This model, named pH-gated, was based on the acquisition of a *gauche*- rotamer for the phosphorylatable His in the phosphatase reaction as a result of a decrease in pH. Our structural and functional experiments addressed this model that have proved to be misleading and so we extended our studies to the other two activities where the phosphorylatable His intervenes (kinase and phosphotransfer). We have tried to include both messages in our title (address the pH-model and extended to all three HK activities), therefore, we would like to maintain the title as it is.

The Abstract. It is not stated which kind of experiments have been made. The only message is that the pH-gated model for the phosphatase activity has been ruled out.

As the reviewer will know, the length of the abstract in Nature Communications is restricted to 150 words. This fact only allows to introduce the problem that has been addressed and the conclusions that have been obtained, making difficult to detail how it has been achieved. Indeed, our previous version exceeded the word limit (165). To adjust to this limit, we have reduced the Abstract trying to state the experiments carried-out as the reviewer suggests.

In page 10 the authors said. In order to answer this question we analyzed if the rotamer could be induced by side-chain interactions or reflected a state of minimum energy. Then which of these two possibilities actually occurs is not clear stated.

The reviewer is right, since we have not calculated the energy state of the rotamer state of the His in our structures we cannot answer this question. Therefore, we have reformulated the sentence in a proper way with the results exposed in the manuscript and now reads. "*In order to answer this question we analyzed the His side chain interactions.*"

We also thank the reviewer for the comment, which has led us to review in more detail our and PDB structural data obtaining interesting observations. We have carried out an analysis of the phosphorylatable His interactions observing that the *gauche*- rotamer is stabilized by the interaction with the main chain of the residue in position -4. This observation is now included in the manuscript as follows (Page 10): "*HK853 structures reported here and the previously deposited in the PDB show that His260 side chain interact with the main chain oxygen of residue at position -4 (A256 in HK853) when adopts the gauche- rotamer. Similar interactions are observed for the phosphorylatable His in the CpxA (PDB 4BIX)⁸, Walk (PDB 5C93)¹⁷, VicK (PDB 4I5S)²¹ and PhoR (PDB 5UKV)²⁰ HKs, supporting that this interaction would be stabilizing the rotamer disposition.*"

Page 12. This argument is not clear

Moreover, at pH 5 the His is a better phosphate donor than at basic pH given the weakening of the P-N bond, counteracting a possible slight loss of nucleophilicity that the Asp could experience when approaching its pKa.

The argument was raised to explain why phosphotransfer activity could be observed at low pH. According to the pKa values for the His (pKa ~6) and Asp (pKa ~3.9), phosphotransfer at pH 5 can take still place as phosphohistidine might be labile

probably due to the propensity of becoming protonated making it a good leaving group whereas Asp will still be in its ionized form to catalyse phosphoryl transfer.

Thus, we have rephrased it for a better understanding and now reads (page 14):
“Moreover, at pH 5 the phospho-His may be a better leaving group than at basic pH given the weakening of the P-N bond due to its propensity to become protonated whereas Asp will still be in its ionized form to catalyse phosphoryl transfer”

In materials and methods. It is not clear if a crystallization screening was performed or the authors went directly to a known crystallization condition.

We apologize for not having explained correctly how the screenings were made. Initial screenings were set up from the crystallization conditions obtained for the complex structures of HK853-RR468 mutant variants published by Podgornaia and co-workers (*Structure* 21, 1636-1647, 2013). Ammonium sulfate was varied along with sodium citrate and Bis-Tris buffered at different pHs. We have clarified this information at Materials and Methods section.

In Figure 6. b. The background corresponding to time 15 at pH 8 is much higher than the others. Why is that? If the different aliquots along time were taken from a single reaction mixture then the background should not differ, should it? c. The lane corresponding to time 0 comes from a different gel. I encourage the repetition of this experiment. Additionally, the same comment as previous figure, why the band at time 0 at pH 5 is less intense than at pH 8.

As the reviewer points out, in Fig 6b, the background of time 15 at pH 8 is higher. In some occasions, trace amounts of radioactive ATP may distort the background signal of lines in the SDS-PAGE gel. As the reviewer has suggested we have repeated the experiment of Figure 6b.

In relation to the intensity of the bands shown at pH 5 time 0 vs pH 8 time 0 at Fig. 6b, it is due to the lower stability of the P-N bond at low pH as we have confirmed by this experiment. To perform the experiment of phospho-stability the residual ATP must be removed after HK autophosphorylation by using gel filtration. This purification step takes around 10-20 min (notice that half-life of P-EnvZ at pH 5 is around 10 min), therefore the starting amount of phospho-protein is reduced around 50-75 % at pH 5 vs pH 8 since we load identical amounts of protein in the gels, explaining the lower intensities for the bands observed at low pH.

Crystallographic tables 2 and 3. Rmeans and CC1/2 (%) values should be added.

We have introduced Rmeans and CC1/2 (%) in the crystallographic tables.

Other minor comments.

In general. The proper way to name the rotamers I believe is trans/gauche- χ_1 rotamer.

We apologize for the slip. We have indicated in the introduction next to the rotamers the χ_1 and highlighted that “(hereafter, χ_1 will be omitted for simplicity)”.

In some the figures. The net charge of sulfate (SO₄²⁻) is missing in the figures.

In the figures, for clarity, we have omitted the net charges of the anion. In the current version we have clarified this point in the figure legends.

In the Results Section. Homogenize subheading with and without stops.

Following the indications of the reviewer we have homogenized subheading by removing the stops.

Page 6. The inactive/active terminology is not totally clear. In this case I believe it corresponds to the phosphatase activity. It should be stated clearly. In the introduction there is a similar situation but there it is unmistakably indicated. (Overall, in the pH-gated model the authors proposed that the transition between inactive to active states for the phosphatase activity ...).

The pH-gated model proposes that in HKs of the HisKA family the rotamer disposition of the phosphorylatable His is regulated by pH, acquiring, an inactive gauche- rotamer at pH between 5.2-6.5 and an active trans rotamer at pHs above 6.5 (Fig 1b).

We have clarified along the manuscript the inactive/active terminology for the rotamer state and the reactions.

Page 6. The fact that this pdb corresponds to the isolated HAMP DHp should be informed in the text. *Finally, contrary to the proposal of the pH gated model the structure crystallized at the lower pH (4.6) corresponding to the isolated EnvZ HAMP DHp.*

We have corrected the text accordingly to the suggestion of the reviewer.

Page 6. I have three comments on this sentence.

Indeed, in those cases where the phosphorylatable His is trapped performing the autophosphorylation reaction, as in subunit A of CpxA (PDB: 4BIW8, solved at pH 8.5) and subunit A of chimeric version of chimeric EnvZ (PDB: 4KP47, solved at pH 7.5), or the phosphatase competent reaction in HK853 (PDB: 3DGE10, solved at pH 5.6), it shows an invariably trans rotamer, that is for an active state. The fact that this pdb corresponds to a chimeric EnvZ should be informed in the text. This sentence seem to be stating the opposite to the previous one. The isomerization of the phosphorylatable His is related with the catalytic state or not? "..., that is for an active state." It is not clear.

We have informed in the text that PDB 4KP4 was a chimeric EnvZ.

In relation to the isomerization and catalytic state of the His, we agree that the isomerization of the phosphorylatable His is related to the catalytic state but independently of the pH. In those structures where there is a conformation competent for a specific reaction, such as the kinase, the His adopts the *trans* rotamer as the active state but the present manuscript shows that the His conformation is not directed by the pH as the pH-gated model proposed.

Page 10. The citation of the previously described stability should be included. Notice that as it was previously described, the phosphorylated RR was stable at the tested pH range (8-5).

The observation of RR~P stability was demonstrated in the experiment described at the previous paragraph where phosphorylation of RR468 was stable at the pH range between 5 and 8. Thus, we just pointed it out in page 10 for the sake of the next experiment. We have clarified this in the text. As the reviewer suggests we have included a citation describing the phospho-Asp stability at low pH (Attwood et al, *Amino Acids*, 2011).

Page 14. This sentence does not belong to the section. Maybe could be moved to Discussion section.

Overall, our in vitro functional assays confirm that, as in the case of HK853, the autophosphorylation and phosphatase reactions are impaired at pH 5 while the phosphotransfer activity is unaffected at this pH, in close relation with the chemistry of the catalytic residues involved in each reaction but not with a side-chain conformational change of the phosphorylatable His as it is proposed in the pH-gated model.

We believe the sentence summarizes all our results presented in the section, so we prefer to leave it where it is.

Figure 1. The atom name labels to the His should be included.

The atom labels have been included

Figure 2. a and c have their zoom in (d and e). b could have zoom in, too.

As the reviewer has suggested we have included the zoom vision of Figure 2b

Figure 4. b. The frame of the pH 5 experiment is thicker. In this case ADP⁻ and ADP⁺ experiments are in different gels. Maybe be it is better to indicate it clearly with a line. So nobody could say it was confusing. For the other pHs (8, 7 and 6), ADP⁻ and ADP⁺ experiments are in the same gel. Is that correct? In the graph there are no error bars in the graph. Why? Is it because the sign of the plot are very big.

For experiment shown at Figure 4b, the phosphatase activity assay performed at each pH (8, 7, 6 and 5) in the absence and presence of ADP was done in the same gel. However, for experiment at pH 5 in the gel showed, the samples were loaded in the wrong order for mistake, thus, we had to rearrange it. However, to avoid confusions we show other gel with the samples loaded properly. In the plot the error bars are now visible.

In the legend of Fig. 1. Two distinct rotamers *trans* and *gauche*- are shown for the catalytic His, where just the *trans* rotamer is involved in the phosphatase reaction.

At Figure 1b when the His is active for the phosphatase reaction adopts the *trans* rotamer, as shown in the structure of the phosphatase-competent complex HK853-RR468 at pH 5.6 (3DGF), thus, we have clarified this in the figure legend. We have also included in Figure 1b the *trans* rotamers for the autophosphorylation and phosphotransfer reactions. In the autophosphorylation reaction, the structures of the chimeric EnvZ (PDB: 4KP4) and CpxA (4BIW) show that the *trans* rotamer is the active state for the His in this reaction. Although we lack of structural information for the phosphotransferase reaction we have included also the *trans* rotamer as the active state of the His for this reaction.

In the legend of Fig. 4. The title is confusing. It would be better in my opinion Phosphatase activity of HK853 on RR468. In b it should be mentioned that the gel corresponds to a representative experiment of the three repetitions. It should be mentioned that the mean values of three independent experiments are plotted in the graph. The bars correspond to the standard deviation.

Following the indications of the reviewer we have changed the title of Fig 4 legend for clarity that now reads as "*RR468 phosphorylation by phosphodonor and dephosphorylation mediated by HK853 at different pHs*". According to the reviewer, we have included in the legend that the gel is a representative experiment of three repetitions and that mean values for the experiment are represented in the graph including error bars.

Crystallographic tables 2 and 3. Total nro. of reflexions. Add unit to the B factor. Average B factor (Å²). PDB code

We have introduced the corrections

Supplemental table S2. The rotameric form of the structures reported here should be checked. They should be all *gauche*-, shouldn't they?

We apologize for the mistake. We have corrected the *trans* for the *gauche*- in the structures of HK853-RR468 at each pH tested included at the Supplemental table S2.

Reviewer #2:

The manuscript by Mideros-Mora et al studies effects of pH on the structure and function of the histidine kinase HK853. By solving and comparing a series of HK853-RR468 complex structures at different pH, the authors argue against a pH-gated model in which the pH determines the *trans*- or *gauche*-position of the phosphorylated His residue and dictates the active or inactive state of HK853. The authors further claim that the pH-dependent activities of HK853 are due to different protonation states of the His residue based on biochemical analyses of various HK activities at different pH. It is not a novel concept that protonation of the His at different pH impacts the enzyme activities. Moreover, biochemical results presented in this manuscript appear not sufficient to favor or disfavor the His protonation model. A pH-dependent conformation-switching model cannot be completely excluded.

The pH-gated model was proposed by Liu and collaborators based in the observation that the phosphoacceptor His of HK853 presented a *gauche*- conformation in the structure of HK853-RR468 complex solved at pH 5.0 while the rotamer conformation of this residue was *trans* in a similar structure reported at pH 5.6. This model was experimentally supported by observing lack or reduction of phosphatase activity in HK853 and EnvZ, respectively, under acidic conditions, and by NMR pH titration experiments for HK853-RR468 systems.

The data presented in the present manuscript completely excluded the original observation that pH induces a conformational change in the phosphoacceptor His. We present structures of the identical complex used for Liu and collaborators to propose the pH-gated model ranging from pH 7.5 to 5.5 and the conformation is always *gauche*-. Our structures solved at high pH show a *gauche*- rotamer when it was supposed to be *trans*, according to the pH-gated model. Furthermore, numerous structures of the PDB demonstrates that the disposition of the His rotamer is completely independent of the pH, including EnvZ, the second HK used by the authors to support enzymatically the model, which shows a *trans* active conformation at a pH as low as pH 4.0 where the model proposed the *gauche*- conformation.

We also demonstrate that the evidence that HK853 lacks of phosphatase activity under acidic conditions is incorrect. In the manuscript we have demonstrated that the phosphatase activity is not completely lost at pH 6 and 5, moreover, in the presence of ADP HK853 retains a considerable phosphatase activity, an effect that was not observed in the pH-gated model as the authors work with a HK853 construct lacking the CA domain. We also confirmed that pH decreases but does not eliminate EnvZ phosphatase activity. In addition, we confirm biochemically that phosphotransferase activity is not or minimal affected by the pH. This observation also rules out the pH-gated model since if the His conformation were modified by the pH it would be no longer properly aligned with the acceptor Asp, as it is proposed in the pH-gated model for the phosphatase reaction, and consequently the activity must be modified in the

same way for both reactions but this is not the case. Oppositely, this is the case for the autophosphorylation reaction where our data shows that the activity decreases similarly to phosphatase. This observation is explainable by the chemistry of the residues working as nucleophile or general base in each reaction (His in kinase and phosphatase, and His-Asp in phosphotransferase) but not for the pH-gated model.

Finally, the authors indicate that the NMR data support the pH model, although the changes observed also could be attributable to the protonation of the His without any conformational change. The authors noticed this possibility in the discussion but they dismissed it because: "*the fast protonation-deprotonation process yields a single set of population-weighted signals as observed in many enzymes*²⁴". However, this observation is not always correct and there are several examples of slow exchange of protons, including classical works (Knoblauch et al, Eur. J. Biochem, 1988), in catalytic His. Therefore, this possibility cannot be ruled out for explaining the NMR observations reported by Lui and collaborators.

All together we honestly believe our results demonstrate that the pH does not influence the disposition of the His rotamer as the pH-gated model proposes, making this model unsustainable.

The following are detailed comments:

1. All of the WT complex structures have two phosphoryl group mimics, a BeF₃ and a sulfate ion, suggesting that both HK and RR proteins are phosphorylated. But none of the kinase, phosphotransfer or phosphatase reactions will result in a complex with both proteins phosphorylated. Are such complexes physiologically relevant? Will the pH-independence of this non-reacting resting state be readily translatable to other relevant structures?

As the reviewer mentions, the structures presented in the manuscript contain two phosphomimetics, BeF₃- and sulfate ion. We do not believe that it corresponds to a physiological state of the HK-RR complex catalyzing a specific reaction, but the structures show the state of the proteins (and the complex) in relevant physiological states. Regarding the structure of the complex, we consider that we are observing the P-RR468 bound to the unphosphorylated HK853 as the structures resemble the complex ascribed for the phosphatase reaction now 10 years ago (Casino et al, *Cell* 2009), but with the His pointing away from the active site in a resting state waiting for the signal to become active and participate at the reaction. The signal is not a change in pH as the structures at pH 5.5, 6.5, 7 and 7.5 show all a similar resting state for the His that we believe is a biological state. Indeed, we have searched at the PDB and found no correlation between the pH and the His conformation at structures of HisKA HKs, showing a similar percentage of structures presenting *gauche-* vs *trans* conformation at acidic or basic pHs. Moreover, some HK structures presented both rotamers in the same dimer. The phosphorylated conformation of RR468 is supported for the structural comparison with individual structures of RR468 in its apo and BeF forms. But, why do we observe a sulfate bound to His if the HK should be dephosphorylated for the phosphatase activity? Our structures support that in the resting *gauche-* rotamer the protein presents a proper environment to accommodate and stabilize the phosphoryl group of the P-His. Therefore, under the crystallization conditions with really high concentration of ammonium sulphate (> 1M), this ion occupies a favorable place but does not indicate that the protein is phosphorylated. This phosphorylatable His in the *gauche-* conformation is a key structural element to generate this proper environment since we cannot observe the sulfate ion once we remove the residue (structures of H260A mutant) even when the concentration of ammonium sulfate in the crystallization conditions is greater than 1 M. In addition, the presence of a sulfate interacting with the His has also been observed in other structures such as HK853 (PDB: 4JAV and 2C2A) and WalK (PDB: 5C93) with the His adopting the *gauche-* conformation. We therefore consider that this structure is

biologically relevant for HKs since it supports that the P-His must be stabilized in this conformation.

2. The main biochemical evidence against the pH-gated switch model is that pH did not affect the phosphotransfer reaction. However, data shown in Fig. 5C did not support the author's claim that phosphotransfer is independent of pH. At pH 5, initial level of HK853~P was less than that at pH 8 but it took a longer time (~15 min) than at pH 8 (1 min) to reach a similar level. This suggests that the reaction rate is lower at pH 5 than at pH 8. Quantification of phosphorylation is required to draw any conclusion about reaction rates.

Following the indications of the reviewer we have quantified the phosphorylation level of the proteins in the different assays. The quantification results support that the pH has minimal or no effect on the phosphotransfer activity in comparison with the high impact on the kinase and phosphatase activities. The quantification shows that at both pHs more than 95% and 85 % of the phospholabelled band has been transferred at the first time point (1 min) for HK853 and EnvZ, respectively. The fact that a small amount of reminiscent P-HK853 is observed was due to the fact that we performed the tests in ratio 1:1 (HK:RR). Quantification of the proteins is not perfect, mainly for RRs with low molecular weight, so there may be some minimal excess of active HK over RR. In the case of pH 8, this is not a problem since HK853 preserves the phosphatase activity and the P-RR is dephosphorylated, so this P-HK excess is eliminated in a new phosphotransfer/phosphatase cycle. In the case of pH 5.0, phosphatase activity is very low, even more when we have eliminated ADP, thus this reminiscent phospholabelled HK takes longer to be removed. In fact, the instability of the P-His at pH 5 ($t_{1/2}$ 10 min) is also contributing to this low rate of eliminating the remaining P-HK853. In the new figure 5c we have repeated the experiments using a HK853:RR468 ratio of 1:1.5 and we have confirmed that almost 98 % of the phosphoryl group is transferred at 1 min, supporting that the phosphotransfer reaction is not affected or affected to a lesser extent than phosphatase and kinase by pH. In any case, it was not our intention to indicate that this reaction is independent of pH. We assume that any reaction involving catalytic residues which protonation state can change within physiological pH is dependent on the pH. However, we believe that our assumption was correct since the phosphotransfer activity is much less affected by pH (at the range 8 to 5) than kinase and phosphatase reactions, supporting that phosphorylatable Asp plays an important role in this reaction being less likely to be influenced by pH 5 due to its low pKa ~3.9. Meanwhile, the autophosphorylation and phosphatase activities are more affected by pH due to its influence on the ionization of the catalytic His. We have clarified this point as follows (Page 14): *"In this way, the phosphotransfer reaction seems to be more independent of pH at the range assayed (from 8 to 5), probably because the hydrolysis of the P-N bond and the phosphoryl transfer is promoted by the attack of the phosphorylatable Asp acidic group which protonation state is weakly affected at pH 5.0 since its pKa is ~3.9. Moreover, at pH 5 the phospho-His may be a better leaving group than at basic pH given the weakening of the P-N bond due to its propensity to become protonated whereas Asp will still be in its ionized form to catalyse phosphoryl transfer."* Therefore, the biochemical data rules out the pH-gated model. If the His conformation is dictated by the pH, as the pH-gated model proposed, lowering the pH forces the His conformational change from *trans* to *gauche*- rotamer, which no longer would be properly aligned with the acceptor Asp, and, consequently, the activity must be modified in the same way for both phosphatase and phosphotransferase reactions but this is not the case.

3. As described above, the kinase, phosphotransfer and phosphatase activities shown in Fig. 4 and 5 are all reduced at acidic pH. These results cannot differentiate a His protonation model or a conformational switch model. pH-dependent conformational

dynamics could occur at other residues and still contribute to the activity difference. The F19 NMR studies in Liu et al suggested the switching of conformation states at a time scale different from the His protonation.

As we have indicated in the previous answer the changes in activity are not similar for the 3 reactions. While the phosphatase and kinase activities are strongly influenced or avoided by lowering the pH, minimal effect is observed in the phosphotransferase activity. We honestly believe that these results distinguish a His protonation model from a conformational switch model. If the pH induces a conformational switch in the His conformation the phosphatase and phosphotransfer activities must be similarly influenced. Our data clearly shows that is not the case. Furthermore, the structural data confirms that pH does not induce a conformational change in the His.

Liu et al support the pH-gated model on the slow exchange time scale for NMR measurements, since they assume: "*the fast protonation-deprotonation process yields a single set of population-weighted signals as observed in many enzymes*²⁴". However, this observation is not always correct and there are several examples of slow exchange of protons, including classical works (Knoblauch et al, Eur. J. Biochem, 1988), in catalytic His.

4. A SO₄ ion can occupy the same position as a PO₃ group. Will the different geometry or position of oxygen atoms in PO₃ allow similar residue interactions as SO₄ to stabilize the gauche position? Is it possible to model a phosphorylated His residue onto the current structure to explore the potential residue interactions?

The geometry of the sulfate bound to the His could differ from the His bound to the phosphoryl group as (i) we are not observing a covalent N-P bond for a phosphorylated HK and (ii) sulfate has four oxygens while the phosphoryl group has three.

Following the reviewer's indication we have generated the HK853 model with His260 phosphorylated in Nε using Vienna-PTM 2.0 server (Margreitter et al, *Nucleic Acid Research*, 2013) and this model was subjected to different cycles of energy minimization with Chimera (Pettersen et al, *J Comput Chem*. 2004). In the figure below, we show how applying minimization cycles, the phosphoryl group is approaching to the sulfate supporting that this position is favorable to accommodate a resting state of the P-His. In the minimized structure the phosphoryl group mediates contacts with HK853 R317 and RR468 K85, two residues that also mediate contacts with the sulfate in some of the reported structures, supporting the phospho-mimetic role of this anion.

Figure. Modelling of phospho-His in the structure of HK853-RR468^{7,5}. a) Structure of the complex at pH 7.5 (in light orange) introducing phospho-His with Vienna-PTM 2.0 server (Margreitter C, et al, *Nucleic Acids Res.* 2013). b) Superposition of structure shown in a) with the same structure after one cycle of minimization (in green). c) Superposition of structures shown in a) and b) with the same structure after three cycles of minimization (in violet). Phosphoryl group approaches sulfate location after cycles of minimization and is stabilized making contacts with R317 (located at the end of $\alpha 2$ in the other subunit of HK853 dimer) as well as with K85 (located in the $L\beta\alpha 4$ in RR468).

5. Fig. 1a, it is generally believed that the His phosphorylation in HKs occurs at N3 position. Highlighting the N3 phosphorylated form may reduce any potential confusion.

We completely agree with the reviewer and we have indicated in Figure 1a that phosphorylation at N3 has been observed at HKs. We have explained in more detail in the figure caption: “Structural and functional studies on HKs support autophosphorylation at $N\epsilon^{7,8,49}$ while autophosphorylation at $N\sigma$ happens at other enzymes such as nucleoside diphosphate kinases⁵⁰”

6. Fig. 4a, at pH 5, the phosphorylation band intensity at 60 min appears much less than corresponding lanes at higher pH values. Does this suggest that the RR autophosphorylation rate is also affected by pH?

We performed each experiment several times (minimum 3 times) and the observed variation between intensities lie within the error of the technique. To avoid confusions, we have used other gel in the figure.

7. Page 11, paragraph 1, the last sentence. “lowering the pH below 6 increases the His protonation...” The wording here imply pH 6 as a threshold value, but neither the protonation mechanism nor the observed data supports the presence of such threshold.

The theoretical pKa for His is 6 (CRC Handbook of Chemistry and Physics, 75th ed.;CRC Press: Boca Roton, FL, 1995) which is in concordance with the *in silico* calculations performed with the program ROSIE server (<http://rosie.rosettacommons.org/>) that proposes a pKa value of 6.1 for the catalytic His in the complex of HK853-RR468. It is very possible that this is not the exact pKa value but we consider that it should not be too far away since a strong decrease in the phosphatase and kinase activities with respect to pH 8 is observed when we approach and low this value, which supports the protonation of the residue.

8. Page 13, paragraph 2, line 9. Typo, “high-life” change to “half-life”.

Thanks for the indication. The typo has been corrected

Reviewer #3:

Altogether, the conclusions are well supported by the presented data and provide convincing evidence that the previously proposed pH-gated model is invalid. This finding is of interest for the community in the field of TCS.

My major concern is that the authors do not sufficiently comment on the discrepancies between their observations and those made by Liu et al regarding the conformational changes they observed at low pH. I have a few minor comments/corrections:

We are gratified that the reviewer finds our manuscript of interest for the community in the field of TCS and thank him/her for the suggestions that will improve the manuscript. We apologize for not describing thoroughly conformational changes or inter-domain arrangements between the structures, but in contrast to Liu et al, our phosphatase-inactive structures at different pHs show a highly similar inter-domain arrangement as the phosphatase-active structure at pH 5.6 (RMSD ~ 0.7Å) and, therefore, a slightly different arrangement from the inactive-phosphatase structure at pH 5 (RMSD ~ 2.0Å) reported by Liu *et al* in the pH-gated model. In that sense, the differences between our structures and Liu et al are similar to those already described by the authors in the pH-gated model (Liu, *Nature Communications*, 2017). They observed that the N-terminal half of the Dhp $\alpha 1$ was less bent (about 2.1 Å) than at the active structure causing movements of the CA domain and the response regulator RR468 around the Dhp which resulted in distance shifts of ~6.9 and ~2.7 Å respectively. As our structures at different pH are similar to the active structure and different from Liu et al, their differences apply to us. Following the reviewer's indication, we have summarized these observations in the manuscript as follows (page 9): "*In the HK-RR^{pHs} structures the Dhp helix $\alpha 1$ and CA domains in HK853 adopt an identical conformation to HK853-RR468^{5.6} and the two molecules of RR468 are also placed close to the Dhp domain as in HK853-RR468^{5.6} structure with no inter-domain rearrangements (Fig. 2b). Thus, slight differences in the relative HK-RR disposition were just observed between HK-RR^{pHs} and HK853-RR468^{5.0} as has been described previously¹⁵ (Fig. 2c)*"

P6, L10: 18 or 20 ?

As the reviewer notices we wrote an incorrect value and the correct one was 20. However, we have introduced four more structures with PDB entries 4CTI, 5UKV, 4MT8 and 6DK7, increasing the number of *gauche*- and *trans* His rotamers to 27 and 39, respectively. Analysis of these 16 extra His confirms the pH independence of the His rotamer.

P2 L8: phosphorylatable.

P3, 2 lines from the bottom: ...similar to the one observed...

P5, L2: ...was based on...

P5, 2 lines from the bottom: the effects

P7, L4: The proposed pH-gated model (ref 15) was structurally...

P9, 12 lines from the bottom: ...phosphorylatable Asp due to its mutation...

P10, L1: ...to the one observed...

P10, L5: ...an equivalent position as HK853.

P10, title: Effect of pH on the phosphatase...

P11, 13 lines from the bottom: ...the in vitro assays support that the...

P13, L11: The pH-gated model was also proposed on the basis of the pH-dependent phosphatase...

P13, 7 lines from the bottom: ...a similar result as in HK853...

P17, L4: EnvZ not in italics

P17, L9: ...were carried out in...
P17, last line: appropriate
P21, L10: HK853
P21, L13: phosphorylatable
P21, 5 lines from the bottom: ...the same code as in a).
P21, 3 lines from the bottom: as sulfate
L22: Fig. 4. RR468 autophosphorylation
L22, last line: delete normalizing
L23, L5: autophosphorylation with
Table 1: what does the * means (His rotamer for each subunit in dimer*)
Legend to Fig. S1, L8: as well as

We thank the reviewer for indicating all these errors that have been corrected in the current manuscript version.

Reviewer #4:

(i) The PDB entry 4CTI (corresponding to a chimera between the HAMP domain of Af1503 and the DHp and CA domains of EnvZ) is missing in Table 1. Some other structures might also be included in the table. For instance, the PDB entries 5UKV, 4MT8 or 6DK7. I guess that the structure of the ThkA-TrrA complex was not included in the analysis because of its low resolution.

We thank the reviewer for reminding us to include other PDB entries. As the reviewer has suggested we have introduced in Table 1 the PDB entries 5UKV (isolated DHp domain of PhoR from *Mycobacterium tuberculosis*), 4MT8 (DHp domain of the hybrid HK ERS1 from *Arabidopsis thaliana*) and 6DK7 (DHp-CA domains of the hybrid HK RetS from *Pseudomonas aeruginosa*). We have also introduced PDB entry 4CTI that corresponds to a chimeric EnvZ. As it can be observed in the Table below, the *gauche*-rotamers can be obtained at pH 7.2 and 7.5 and *trans* rotamers can be obtained at pH 7.5 and 4.0. In the case of RetS *gauche*- and *trans* rotamers are observed at the same pH, confirming our previous observations that rotamer conformation is pH independent and ruling out the pH-gated model.

PDB	Protein	pH	His rotamer for each subunit in dimer	Crystallization mother liquor
4CTI	EnvZ (HAMP _{Af1503})	4.0	trans/trans/trans/trans	20 % PEG 3350, 0.2 M Lithium acetate, 0,1M MMT buffer pH 4.0
5UKV	PhoR ^{DHp}	7.2	gauche-/gauche-	35% PEG 200, 2 mM EDTA, 0.2 M KI, 0.1M Na/K phosphate pH 7.2
4MT8	ERS1 ^{DHp}	7.5	gauche-/trans	9% PEG 3350, 0.18 M l-proline, 0.1 M HEPES pH 7.5
6DK7	RetS ^{DHp-CA}	7.5	trans/trans/trans/trans /gauche-/gauche- /gauche-/gauche-	2.7 M NaCl, 9mM CoCl ₂ , 90 mM HEPES pH 7.5

As the reviewer indicates we have not included the structure of ThkA-TrrA complex for its low resolution.

(ii) Various of the papers reporting the crystallization conditions shown in Table 1 are not cited in the manuscript.

We have introduced the references for the described crystallization conditions of Table 1. Below it can be seen the references added.

EnvZ HAMP-DHp: Ferris HU et al, *Structure* 2012 (PDBs 3ZRV, 3ZRX and 3ZRW)

EnvZ-Af1503 HAMP: Ferris HU et al, *J Struct Biol* 2014 (PDB 4CTI)
Vick: Wang C et al *Plos Biol* 2013 (PDB 4I5S)
EnvZ DHp: Eguchi Y et al *J. Antibiot* 2017 (PDBs 5B1N and 5B1O)
CpxA: Mechaly AE et al *Structure* 2017 (PDB 5LFX)
PhoR DHp: Xing D et al *ACS Omega* 2017 (PDB 5UKV)
ERS1 DHp: Mayerhofer H and Mueller-Dieckmann J, *J Biol Chem* 2014 (PDB 4MT8)
RetS DHp-CA: Mancl JM and Schubot FD, *Structure* 2019 (PDB 6DK7)

(iii) CC1/2 values should be reported in Tables 2 and 3.

We have introduced CC1/2 values at crystallographic tables.

(iv) The authors suggest that some of the structures represent a resting state prior either phosphotransfer or dephosphorylation. I would also consider another possibility, a snapshot right after the dephosphorylation reaction, with the sulfate ion mimicking the leaving Pi molecule.

We thank the reviewer for proposing this tempting idea. However, that may indicate that upon dephosphorylation the phosphoryl group returns back to the His, as in a reversible phosphotransfer step which does not seem to be present in HKs but in some phosphorelay systems. The phosphatase activity is thought to proceed through the action of a water molecule to catalyse phosphoryl hydrolysis, and the His just seems to act as a general base and should not hold the phosphate group.

We believe that the resting state is a biological state of the His before any reaction can take place. The His could also acquire this state upon autophosphorylation awaiting the RR in its apo state for the phosphotransfer. The presence of the sulfate interacting with the His, also observed in other structures, has led us to propose that hypothesis.

(v) Figure 4. I assume that the phosphatase assays were carried out using radioactive AcP. In that case, it should be mentioned in the Results and Methods sections.

We have included in the manuscript the use of radioactive AcP

(vi) Did the authors perform phosphotransfer assays like those shown in Figure 5c, using the phosphatase-defective mutant T264A instead of wild-type HK853? A priori using this mutant the phosphotransfer assays wouldn't be affected by the strong phosphatase activity of HK853.

We thank the reviewer for his suggestion. In the current version of the manuscript we have performed new phosphotransfer experiments with HK853 using a ratio (HK853:RR468) 1:1.5 demonstrating that > 98% phosphotransfer occurs in the first time point (1 min). Possibly, accuracy in protein quantification together with differences in the equilibrium between phosphotransfer/phosphatase activities could have been behind the presence of a reminiscent band of P-HK in the phosphotransfer experiments. The use of mutant T264A for HK853 in the phosphotransfer reaction is an attractive idea, however, as this residue plays a key role in the phosphatase activity, in order to avoid any possible influence that this residue may have in the environment of the phosphotransfer reaction, we have preferred to do the experiments with the wild-type proteins.

Reviewers' comments:

Reviewer #1 (Remarks to the Author):

Response to the re-submitted manuscript

In my opinion, the authors have responded robustly to all the reviewers' questions and have changed the manuscript accordingly. The quality of the manuscript has substantially been improved and is now appropriate for publication.

Finally, I will add two minor comments.

In page 14, line 336, I would change "more independent" for "less dependent"

I would include these sentences in MS, which the authors wrote in response to reviewer 2: In addition, we confirm biochemically that phosphotransferase activity is not or minimal affected by the pH. This observation also rules out the pH gated model since if the His conformation were modified by the pH it would be no longer properly aligned with the acceptor Asp, as it is proposed in the pH-gated model for the phosphatase reaction, and consequently the activity must be modified in the same way for both reactions but this is not the case. Oppositely, this is the case for the autophosphorylation reaction where our data shows that the activity decreases similarly to phosphatase. This observation is explainable by the chemistry of the residues working as nucleophile or general base in each reaction (His in kinase and phosphatase, and His-Asp in phosphotransferase) but not for the pH-gated model.

Reviewer #2 (Remarks to the Author):

The revised manuscript presented convincing arguments that the rotamer positions of the catalytic His are independent of pH. Even though the solved structures represent an inactive resting state not corresponding to any catalytic states, the manuscript provided sound reasoning that the observed pH-independent rotamer conformation is relevant, arguing against the previous pH-gated model.

The revisions addressed some of my concerns about the previous version, however, to my opinion, the biochemical analyses are not sufficient for conclusion of the phosphotransferase activity being less dependent on pH than the phosphatase or kinase activity.

New data presented in Fig. 5c with a HK:RR ratio of 1:1.5 addressed my previous comments about phosphotransferase activity. However, the new data with inadequate time range are not sufficient to support pH independence or less dependence. Phosphotransfer appeared completed at the first time point at 1 min, thus the natural conclusion would be that the reactions are too fast to differentiate the activities at pH 5 and 8. For example, phosphotransfer could be pH dependent with the half times of 1 s at pH 8 and 10 s at pH 5, but still result in nearly complete transfer at 1 min as shown here. Kinetic data at early time points, e.g. 30 s or 10 s, showing intermediate phosphotransfer levels, are required for proper comparison. If intermediate phosphotransfer levels cannot be achieved at the earliest possible time, conditions that slow down the reaction, such as temperature decrease, may be tried.

Phosphotransfer data for EnvZ-OmpR in Fig. 6c are inconclusive as well. More EnvZ~P remained at pH5 than at pH8, is this due to less phosphotransfer or excess EnvZ~P? What is the ratio of HK:RR? Again, data at early time points with intermediate phosphotransfer levels are required for proper comparison. For both Fig. 5c and 6c, quantification curves should be focused around

intermediate phosphotransfer levels within the time range of 1 min, it is very hard to compare data from two pH at the current scale.

Reviewers' comments:

Reviewer #1:

In my opinion, the authors have responded robustly to all the reviewers' questions and have changed the manuscript accordingly. The quality of the manuscript has substantially been improved and is now appropriate for publication.

We thank the reviewer for the comment.

Finally, I will add two minor comments.

In page 14, line 336, I would change "more independent" for "less dependent"

Following the indication of the reviewer we have changed the expression.

I would include these sentences in MS, which the authors wrote in response to reviewer 2:

In addition, we confirm biochemically that phosphotransferase activity is not or minimal affected by the pH. This observation also rules out the pH gated model since if the His conformation were modified by the pH it would be no longer properly aligned with the acceptor Asp, as it is proposed in the pH-gated model for the phosphatase reaction, and consequently the activity must be modified in the same way for both reactions but this is not the case. Oppositely, this is the case for the autophosphorylation reaction where our data shows that the activity decreases similarly to phosphatase. This observation is explainable by the chemistry of the residues working as nucleophile or general base in each reaction (His in kinase and phosphatase, and His-Asp in phosphotransferase) but not for the pH-gated model.

Following the suggestion, we have included this sentence in the discussion of the manuscript.

Reviewer #2:

The revised manuscript presented convincing arguments that the rotamer positions of the catalytic His are independent of pH. Even though the solved structures represent an inactive resting state not corresponding to any catalytic states, the manuscript provided sound reasoning that the observed pH-independent rotamer conformation is relevant, arguing against the previous pH-gated model.

We thank the reviewer for the comment.

The revisions addressed some of my concerns about the previous version, however, to my opinion, the biochemical analyses are not sufficient for conclusion of the phosphotransferase activity being less dependent on pH than the phosphatase or kinase activity.

We honestly believed that the manuscript provides evidence that the phosphotransfer reaction is less affected by pH than the kinase and phosphatase reactions which activity decrease in a higher extent than the former, at least in the time range tested. It has never been our intention to indicate that the pH cannot have an effect on the phosphotransference reaction, but the magnitude of this effect is different in each reaction according to the pKa of the residue that shows greater catalytic contribution. Thus, when similar times are tested, the phosphotransference activity, where the Asp residue plays a major role, is less affected than the autophosphorylation and dephosphorylation reactions where the His is the main player. In any case, the major

point of the manuscript, which is reflected in the title, is the response to the pH-gated mechanism previously published in Nature Communications. Our manuscript proves that this mechanism is wrong, and therefore warns the scientific community on possible interpretations based on this mechanism.

New data presented in Fig. 5c with a HK:RR ratio of 1:1.5 addressed my previous comments about phosphotransferase activity. However, the new data with inadequate time range are not sufficient to support pH independence or less dependence. Phosphotransfer appeared completed at the first time point at 1 min, thus the natural conclusion would be that the reactions are too fast to differentiate the activities at pH 5 and 8. For example, phosphotransfer could be pH dependent with the half times of 1 s at pH 8 and 10 s at pH 5, but still result in nearly complete transfer at 1 min as shown here. Kinetic data at early time points, e.g. 30 s or 10 s, showing intermediate phosphotransfer levels, are required for proper comparison. If intermediate phosphotransfer levels cannot be achieved at the earliest possible time, conditions that slow down the reaction, such as temperature decrease, may be tried.

As the reviewer suggests, it is possible that within the time range tested in the manuscript (1 to 30 min) cannot be distinguished the dependence of the pH for the phosphotransfer reaction at shorter times (< 1 min). However, measuring phosphotransferase activity at really short times (1-30 s) using the technical approach followed in the manuscript, which is the most widely used in the field, could introduce a lot of error, making difficult to evaluate whether the differences (if were observed) are due to pH or due to experimental causes. Therefore, it would be necessary to develop stop-flow techniques to address this question. As this is not an essential point of the manuscript and as we cannot ensure completely its independency of pH at shorter time range < 1 min, we have decided to include the reviewer concerns in the manuscript related with the pH dependence at a shorter time range indicating that (page 14) "Although we could not discard pH dependency for phosphotransferase reaction at shorter times (< 1 min)". In addition, each time the biochemical results for the three reactions are compared, it is specified that this comparison is "at the time range tested".

Phosphotransfer data for EnvZ-OmpR in Fig. 6c are inconclusive as well. More EnvZ~P remained at pH5 than at pH8, is this due to less phosphotransfer or excess EnvZ~P? What is the ratio of HK:RR? Again, data at early time points with intermediate phosphotransfer levels are required for proper comparison. For both Fig. 5c and 6c, quantification curves should be focused around intermediate phosphotransfer levels within the time range of 1 min, it is very hard to compare data from two pH at the current scale.

The ratio HK:RR in all phosphotransfer assays, either for HK853:RR468 or EnvZ:OmpR was 1:1.5. We are sorry for not indicating it in the previous version for EnvZ:OmpR, thus, it has been indicated in the current one.

As mentioned in the previous point, and in agreement with the reviewer concerns about the dependency of the phosphotransfer reaction with pH within a shorter time range (< 1 min), we have clarified in the manuscript that the less dependency of the pH for the phosphotransfer activity versus kinase and phosphatase is at the time range tested, and we have added the following sentence to indicate that dependency of pH at shorter time range (< 1 min) cannot be discarded (page 15) "although we could not dismiss pH dependency at shorter times (< 1 min)"